# Securing Transfer Learning: Few-Shot Training-Data Reconstruction Attacks and Reverse-Role Homomorphic Encryption

## Abstract

The growing body of literature on training-data reconstruction attacks raises significant concerns about deploying neural network classifiers trained on sensitive data. However, differentially private (DP) training (e.g. using DP-SGD) can defend against such attacks with large training datasets causing only minimal loss of network utility. Folklore, heuristics, and (albeit pessimistic) DP bounds suggest this fails for networks trained with small per-class datasets, yet to the best of our knowledge the literature offers no compelling evidence. We directly demonstrate this vulnerability by significantly extending reconstruction attack capabilities under a realistic adversary threat model for few-shot transfer learned image classifiers. We design new white-box and black-box attacks and find that DP-SGD is unable to defend against these without significant classifier utility loss. To address this, we propose a novel homomorphic encryption (HE) method that protects training data without degrading model's accuracy. Conventional HE secures model's input data and requires costly homomorphic implementation of the entire classifier. In contrast, our new scheme is computationally efficient and protects training data rather than input data. This is achieved by means of a simple role-reversal where classifier input data is unencrypted but transfer-learned weights are encrypted. Classifier outputs remain encrypted, thus preventing both white-box and black-box (and any other) training-data reconstruction attacks. Under this new scheme only a trusted party with a private decryption key can obtain the classifier class decisions.

## 1 Introduction

Since neural networks retain imprints of their training data Arpit et al. (2017); Shokri et al. (2017); Carlini et al. (2019); Song & Shmatikov (2020) it is crucial to train them using privacy-preserving methods - especially in applications where the training-data contains highly sensitive information. Governments have recognised training-data privacy risks as a central issue in the development of ML/AI systems and mention membership inference Shokri et al. (2017) and model inversion Fredrikson et al. (2014; 2015) attacks (revealing queried examples or training-data 'prototypes'/close representatives) explicitly in their official documents Council of European Union and European Parliament (2024); UK Department for Science, Innovation and Technology (2025).

Differential privacy (DP) Ponomareva et al. (2023) is the de facto method for privacy-preserving neural-network training. DP provides formal privacy guarantees quantified via the *privacy budget*, $\epsilon \geq 0$, and the primary DP algorithm for machine learning purposes is Differentially Private Stochastic Gradient Descent DP-SGD Abadi et al. (2016). Smaller $\epsilon$ guarantees higher privacy which is achieved by injecting more noise into the DP-SGD training algorithm, degrading neural network utility in turn. In its original database setting, this privacy–utility tradeoff of DP is more pronounced for small datasets (Dwork & Roth, 2014). A similar effect might reasonably be expected for DP-SGD, i.e severe degradation in neural network performance when used to secure small training-sets; indeed Abadi et al. (2016); Tramer & Boneh (2021) discuss this effect. This can be quantified by the *excess empirical risk* (Bassily et al., 2014), but current theory covers only convex loss functions and asymptotically large training sets. Precisely quantifying this behavior for DP-SGD under realistic threat models remains a challenging open problem. Current privacy accounting theory assumes that an adversary has access to the neural network parameters and the intermediate gradients during training.

A more realistic *hidden state threat model* (Feldman et al., 2018; Balle et al., 2019; Ye & Shokri, 2022; Altschuler & Talwar, 2022; Cebere et al., 2025) assumes no access to the intermediate gradients, but derivation of tight privacy-budget bounds for this model is currently intractable (Altschuler & Talwar, 2022; Cebere et al., 2025). The seminal work of (Balle et al., 2022) not only assumes adversary's access to gradients but also full knowledge of all training-data instances bar one. Given that very strong assumption (the *informed adversary*) a major contribution of Balle et al. (2022) is to show how the difficulty of recovering the missing data-item relates to the DP-SGD privacy budget.

In the hidden state threat model the above results are pessimistic – far less privacy may suffice to defend against a data reconstruction attack. Lacking relevant theory, security advisors may push overly damaging defenses "just to be on the safe side" prompting user doubt about their practical value. In some cases meeting such pessimistic security bounds can cripple utility sometimes leaving neural networks unusable or outperformed by simpler methods (Tramer & Boneh, 2021). We would like to dispel such skepticism where it is misplaced and can lead to dangerously unsecured networks. In all cases where DP-SGD is genuinely incapable of providing an acceptable privacy-utility tradeoff, we should provide new alternative defense methods. As a first step, we provide firm evidence of the need for alternative security measures in few-shot transfer learning (TL) which is widely used in practice due to its effectiveness and computational efficiency (Chen et al., 2019b; Kolesnikov et al., 2020). TL and few-shot TL (and even one-shot) are commonplace in sensitive domains where labeled data are scarce: e.g., medical imaging (small numbers of labeled scans per institution) (Pachetti & Colantonio, 2024), network intrusion detection using sensitive network logs, biometrics and facial expression recognition, and financial fraud detection using transaction records, see Song et al. (2023). Working in a realistic threat model where, as discussed above, theoretical analysis is currently unavailable, we design sophisticated and bespoke training-data reconstruction attacks that are powerful enough to operate under this threat model. We show that image classifiers transfer-learned on datasets containing few examples per class can suffer accuracy degradation of $10 - 30$ percentage points when trained with DP-SGD strong enough to defend against our data reconstruction attacks. This setting is also common in ML problems with class-imbalanced data Chan & Stolfo (1998); Radivojac et al. (2004); Van Horn et al. (2018), where downsampling/reweighting Ren et al. (2018); Bankes et al. (2024) effectively creates few-shot training sets.

**Our Threat Model (Weak Adversary)**   The adversary has access to the released model and their aim is to reconstruct some examples from the training dataset. Note that successful reconstruction of just a single data item should be considered a serious privacy breach. More specifically, we assume the following threat model in which (only) the following hold.

(A.1)  White-box: model's architecture and released parameters $\theta$ are known.

(A.1*)  Black-box: model's architecture and its hard-label decisions for any input data are known.

(A.2)  Model's training algorithm $\mathcal{A}$ is known.

(A.3)  Prior distribution, $\pi$, from which the training data has been sampled and the size of the training set $N$ are known.

We call an adversary satisfying only (A.1/1*–A.3) the *weak adversary*, in contrast to the *informed adversary* that has been considered in previous works Balle et al. (2022); Hayes et al. (2023). Assumptions (A.1/1*–A.3) effectively mean that the trained model (but not the transfer-learned weights in case of the black-box attack) and its transfer-learning code are public. The adversary knows all the hyper-parameters used in the transfer-learning (number of epochs, learning rate, weight decay, batch size, the initialization distribution of the weights, etc.), *but not* the gradients that were computed during the training or the random seeds used for weight initialization/minibatch sampling. This is indeed realistic – models are often publicly released, while seeds and intermediate gradients are not. Even if the above details are kept private, "*privacy (and security) through obscurity is generally regarded as a bad practice.*" (Balle et al., 2022).

Whilst we confirm that few-shot TL is vulnerable to our training-data reconstruction attacks against which DP-SGD cannot defend, we also find that TL is very well-suited to a novel, efficient defense based on homomorphic encryption (HE, see Section 4). This defense also blocks membership and property inference attacks; the former is even harder than reconstruction to defend with DP-SGD, and the latter is known to be undefendable with DP-SGD when properties are global (Ganju et al., 2018). Any adversary that reconstructs private data under HE must be capable of breaking the encryption scheme which has been robustly scrutinized by cryptographic experts and costed as involving at least

$2^{128}$ computer operations which is well beyond feasibility of compute power in the foreseeable future (including quantum computing) (Hales, 2024).

**Our main contributions are as follows:**

1. We substantially extend the capabilities of white-box attacks and introduce novel highly effective hard-label black-box attacks, giving direct evidence, under realistic assumptions for the adversary, that few-shot transfer learning is vulnerable to such attacks (Section 3).

2. We demonstrate that these attacks cannot be defended against by DP-SGD without severe classifier accuracy degradation (Section 3.2).

3. We introduce a principled Neyman–Pearson scheme for evaluation of the effectiveness of a reconstruction attack that accounts for both false positives and false negatives, improving on previous approaches (Section 2).

4. We devise the Reverse-Role Homomorphic Encryption (RHE) scheme, defending training data in transfer-learning against any reconstruction, membership or property-inference attack (Section 4).

5. We show that this RHE defence does not degrade classifier utility and, despite using homomorphic encryption, provides a practical level of classifier efficiency (Section 4.2).

The novelty of our hard-label black-box attack relies on combining our bespoke white-box attack with the black-box weight extraction (Carlini et al., 2025) resulting with a first attack that produces faithful high quality reconstructions (see Table 2 and Figure 20) and is robust against DP-SGD.

**Comparison with Prior Work**    Despite recent significant progress in (white-box) training data reconstruction attacks, some are defendable by DP-SGD and none of the existing approaches applies to our weak adversary threat model. Appendix A analyzes these and prior black-box attacks, reviews implementations of HE in TL, and discusses connections and key differences between our work vs. model inversion (studying *exact/faithful* reconstruction vs. representative sampling).

## 2 RECONSTRUCTION ROBUSTNESS MEASURES

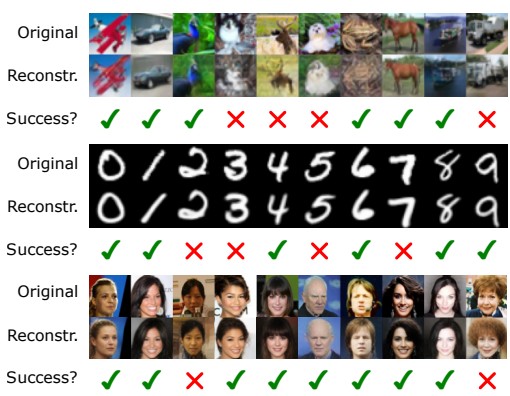

Figure 1: Reconstructions obtained via our attack with CIFAR-10 (Krizhevsky, 2009), MNIST (Deng, 2012) (reconstructed in the $32 \times 32$-resolution) and CelebA (Liu et al., 2015) (reconstructed in the $64 \times 64$-resolution). The reconstruction "success" is determined via the Neyman-Pearson criterion at reconstruction $FPR = 1\%$, see Section 2. TL training set size $N = 10$.

In order to evaluate the effectiveness of a reconstruction attack (or a model's robustness against these attacks) one needs to report both true- and false-positive reconstruction rates. We use ROC and $TPR$-at-low-$FPR$ evaluation based on the preceptual LPIPS metric (Zhang et al., 2018) with appropriate FPR thresholds, instead of naive metrics (such as MSE used in prior work) or visual inspection of reconstruction examples that can overstate success. These stricter metrics are more informative for practitioners evaluating deployed TL systems.

**Notation**    Let $\mathcal{Z} \subset \mathbb{R}^d$ be the domain where the training data lives. We denote by $\pi$ be the prior distribution of the training data over $\mathcal{Z}$ and by $\pi_N$ the product-prior over $\mathcal{Z}^N$. The (normalized) reconstruction error function will be denoted by $\ell : \mathcal{Z} \times \mathcal{Z} \to [0, 1]$. The weights of the neural network model will be denoted by $\theta \in \Theta$, where $\Theta$ is the range of a randomized training mechanism $M : \mathcal{Z}^N \to \Theta$. A reconstruction attack $\mathcal{R} : \Theta \to \mathcal{Z}$ deterministically maps weights of the trained model to a reconstruction.

**The Hypothesis Testing Approach**    We measure effectiveness of a reconstruction attack with respect to a given training set $\mathcal{D}_N = \{Z_1, \ldots, Z_N\}$ by calculating the min-error function $\ell_{\min} = \min_{Z \in \mathcal{D}_N} \ell(Z, \mathcal{R}(\theta))$. The training mechanism is random and the training data is drawn randomly form $\pi_N$, thus $\ell_{\min}$ is also a random variable. We

look at the distribution of $\ell_{\min}$ from the hypothesis testing perspective where we aim to distinguish between the following two hypotheses. *The null hypothesis $H_0$ states that the reconstruction $\mathcal{R}(\theta)$ comes from weights $\theta$ of a model trained on $\mathcal{D}_N$ i.e.,*

$$H_0: \quad \ell_{\min} = \min_{Z \in \mathcal{D}_N} \ell(Z, \mathcal{R}(\theta)) \quad \text{with} \quad \mathcal{D}_N \sim \pi_N, \, \theta \sim M(\mathcal{D}_N).$$

We denote the resulting probability density of $\ell_{\min}$ as $\rho_0$. *The alternative hypothesis $H_1$ states that $\theta$ are the weights of a model trained on a different training set $\mathcal{D}'_N$* i.e.,

$$H_1: \quad \ell_{\min} = \min_{Z \in \mathcal{D}_N} \ell(Z, \mathcal{R}(\theta)) \quad \text{with} \quad \mathcal{D}_N \sim \pi_N, \, \mathcal{D}'_N \sim \pi_N, \, \theta \sim M(\mathcal{D}'_N).$$

We denote the resulting probability density of $\ell_{\min}$ as $\rho_1$. When deciding between $H_0$ and $H_1$, the Neyman-Pearson hypothesis testing criterion accepts $H_0$ for a given value of $\ell_{\min}$ when the likelihood-ratio satisfies $\rho_0(\ell_{\min})/\rho_1(\ell_{\min}) > C$ for a given threshold value $C \in \mathbb{R}_{\geq 0}$. The Neyman-Pearson *true-positive rate* ($TPR_{NP}$) and the *false-positive rate* ($FPR_{NP}$) (both are functions of $C$) are defined as the probabilities of accepting $H_0$ under $\ell_{\min} \sim \rho_0$ and $\ell_{\min} \sim \rho_1$ respectively. Intuitively, $FPR$ measures how good the attack $\mathcal{R}$ is at generating random images acting as likely reconstructions. Thus, from the perspective of the adversary it is desirable to achieve high $TPR$ at low $FPR$, while a defender may require that this $TPR$-at-low-$FPR$ falls below a certain acceptable threshold. The *ROC curve* shows how $TPR$ changes as a function of $FPR$ when varying $C$. One might reliably quantify the attack by finding the threshold $C$ for which $FPR = 1\%$ or $FPR = 0.1\%$ and calculating the corresponding $TPR$. This fact has also been emphasized in the context of MIA (Carlini et al., 2022; Kulynych et al., 2024). The Neyman-Pearson Lemma (Neyman et al., 1933) asserts that the so-obtained ROC curve describes the uniformly most powerful hypothesis test. Note that in practice we typically cannot exactly evaluate the densities $\rho_0$ and $\rho_1$. However, as we show experimentally in Appendix F, for our attacks the random variable $\phi = \log(\ell_{\min}/(1 - \ell_{max}))$ is approximately normally distributed when $\ell \in [0,1]$. This allows one to apply the Neyman-Person criterion using the estimated Gaussian probability density of $\phi$. One can also consider the *cumulative TPR/FPR* defined as $TPR_{cum}(\tau) := \mathcal{P}_{\ell_{\min} \sim \rho_0}[\ell_{\min} < \tau]$, $FPR_{cum}(\tau) := \mathcal{P}_{\ell_{\min} \sim \rho_1}[\ell_{\min} < \tau]$. These are more straightforward to estimate, albeit the resulting (cumulative) ROC curves can be sub-optimal. In our experiments we have observed that the Neyman-Person and the cumulative ROC curves were extremely close to each other (see Appendix F.1) . Theorem 1 (proved in Appendix F) shows that this holds universally at low $FPR$.

**Theorem 1.** Assume that there exists a strictly increasing differentiable transformation $\Phi$ such that $\Phi(\ell_{\min}) \sim \mathcal{N}(\mu_0, \sigma_0^2)$ if $\ell_{\min} \sim \rho_0$ and $\Phi(\ell_{\min}) \sim \mathcal{N}(\mu_1, \sigma_1^2)$ if $\ell_{\min} \sim \rho_1$ with $\mu_0 < \mu_1$. Then, the Neyman-Pearson ROC is asymptotically equivalent to the cumulative ROC at low $FPR$.

Many prior works evaluate the data reconstruction attacks using only $TPR_{cum}$ – see *reconstruction robustness* in Balle et al. (2022); Hayes et al. (2023). In the context of this work this becomes uninformative for large $N$: a trivial attack achieves $TPR \to 1$ (see Lemma 1 in Appendix F). In Appendix F.1 we also compare our attack to a baseline that simply draws the reconstructions randomly from the prior $\pi$. These ideas are further formalized via a modified membership-inference security game (Carlini et al., 2022) (Appendix F).

**Selecting the error function $\ell$**   A common choice for the error function $\ell$ is the mean-squared error ($MSE$) (Balle et al., 2022; Hayes et al., 2023). However, $MSE$ often fails to capture structural features of the images when compared to other perceptual metrics such as SSIM (Wang et al., 2004) or LPIPS (Zhang et al., 2018). In our evaluation methodology we use the LPIPS error function since it turned out to produce best ROC curves (it is also a component of the loss function in our reconstruction attack, see Section 3.1). $TPR$-at-low-$FPR$ results are reported in Table 2 and the ROC curves are reported in Appendix F.

## 3   WHITE-BOX AND BLACK-BOX ATTACKS AND THEIR ROBUSTNESS

**Transfer Learning Setup**   In this work, we target neural network image classifiers trained via transfer learning (TL) (Olivas et al., 2009; Chen et al., 2019b). We consider the TL where: 1) a base model is pretrained for a general classification task on a large dataset and 2) the pretrained model is subsequently adapted to a new task where the training dataset is small by replacing its output layer with a fully connected *head neural net*. We train only the new head-NN with the remaining

| TL Task | Base Model | Base Task | Head Size |
|---------|------------|-----------|-----------|
| MNIST (10-class) | VGG-11 | EMNIST-Letters | 10 (WB) or $10 - 10$ (BB) |
| CIFAR-10 (10-class) | EfficientNet-B0 | CIFAR-100 | 10 (WB) or $16 - 10$ (BB) |
| CelebA (binary) | WideResNet-50 | ImageNet-1K | $4 - 1$ (WB and BB) |

Table 1: TL setups. Each of the base models (Simonyan & Zisserman, 2015; Tan & Le, 2019; Zagoruyko & Komodakis, 2016) has been pretrained on the corresponding base task: EMNIST-Letters (Cohen et al., 2017), CIFAR-100 or ImageNet-1K (Deng et al., 2009). The transfer-learning dataset sizes in the experiments range from $N = 10$ to $N = 40$. In the MNIST and CIFAR-10 experiments the classes were balanced (i.e. $M$-shot learning with $M \in \{1, 4\}$) while in the CelebA experiment we emulate transfer learning for binary face recognition with unbalanced data – the positive class constituted $10\%$ of the training set. We use the cross-entropy as the training loss (with class reweighing in the unbalanced case). The head-NN size $d_1 - d_2$ denotes the architecture $In \rightarrow FC(d_1) \xrightarrow{ReLU} FC(d_2) \rightarrow Out$ where $FC(d)$ is the fully connected layer of width $d$. Different head-NN architectures were used for the white-box (WB) and black-box (BB) attacks.

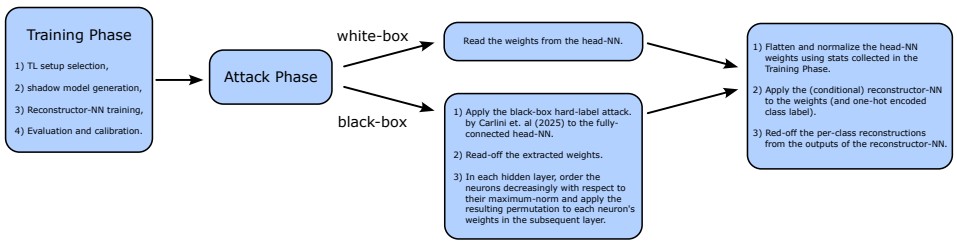

Figure 2: Flowchart summarizing the black-box and white-box attack process. The flowchart summarizing the training phase of the reconstructor-NN is presented in Figure 13 in Appendix G.

weights of the base model being frozen. Thus, the head-NN learns using pretrained deep features. Our weak-adversary assumptions (A.1/1*–A.3) apply to the small TL dataset and training procedure – we do not reconstruct the pretraining data. We emulate this setup in experiments described in Table 1. Care was taken to ensure that the transferred models attained high accuracy - see Appendix I.2.

## 3.1 RECONSTRUCTION ATTACK RESULTS AND METHODOLOGY

We demonstrate that in the TL setup our white-box attack reconstructs some of the training data-points from the one-shot transfer-learning dataset with reconstruction $TPR$ between $50 - 80\%$ at $FPR = 1\%$, depending on the experiment (see Table 2 and examples in Fig. 1). In four-shot learning the $TPR$ values are lower (between $5.5 - 27\%$ at $FPR = 1\%$), but the attack is still effective and reconstructions are of high quality. In Figure 2 we summarize the entire attack process in a flowchart.

**Reconstructor NN and the Shadow Training Method** The reconstructor NN is trained using the shadow model training method introduced by Shokri et al. (2017) that relies on creating a large number of "shadow models" that mimic the model to be attacked, see Algorithm 1. We split each

| Experiment | $N = 10$ | | | | $N = 40$ | |
|------------|----------|------|----------|------|----------|------|
| | $TPR@FPR = 1\%$ | | $TPR@FPR = 0.1\%$ | | $TPR@FPR = 1\%$ | $TPR@FPR = 0.1\%$ |
| | WB | BB | WB | BB | WB | WB |
| MNIST | $49.8\%$ | $29.3\%$ | $15.4\%$ | $7.1\%$ | $5.5\%$ | $1.1\%$ |
| CIFAR | $62.4\%$ | $35.4\%$ | $24.3\%$ | $9.8\%$ | $12.9\%$ | $2.9\%$ |
| CelebA | $80.2\%$ | $68.6\%$ | $49.6\%$ | $48.3\%$ | $27.1\%$ | $9.9\%$ |

Table 2: Reconstruction rates for the white-box (WB) and black-box (BB) attacks for the MNIST, CIFAR-10 and CelebA experiments depending on the training set size $N$ of the attacked model.

---

**Algorithm 1:** Shadow model training

---

**Input:** Dataset $\mathcal{D}$ sampled from the prior $\pi$, classifier training set size $N$, weight initialization
        distribution $\pi_\Theta$, transfer-learning algorithm $\mathcal{A}$, number of shadow models $N_{shadow}$.
**Output:** Shadow dataset $\mathcal{D}_{shadow}$.
Initialize $\mathcal{D}_{shadow} \leftarrow [\,]$ as an empty list.
**for** $k \leftarrow 1$ **to** $N_{shadow}$ **do**
    |   Sample the training dataset $\mathcal{D}_k$ from $\mathcal{D}$, with size $|\mathcal{D}_k| = N$.
    |   Initialize the weights and biases of the trainable layers $\theta_0 \sim \pi_\Theta$.
    |   Train the trainable layers $(\theta_0, \mathcal{D}_k) \xrightarrow{\mathcal{A}} \theta_k$.
    |   Append to the list: $\mathcal{D}_{shadow} \leftarrow \mathcal{D}_{shadow} + [(\mathcal{D}_k, \theta_k)]$.

---

dataset $\mathcal{D}$ (MNIST, CIFAR-10, CelebA) into disjoint $\mathcal{D}_{train}$ and $\mathcal{D}_{val}$, run Algorithm 1 on each
to obtain the disjoint shadow datasets $\mathcal{D}_{shadow}^{train}$ and $\mathcal{D}_{shadow}^{val}$. The reconstructor NN is trained on
$\mathcal{D}_{shadow}^{train}$ and tested on $\mathcal{D}_{shadow}^{val}$. It takes the flattened weights/biases of the attacked head-NN's and
outputs a single image. In MNIST and CIFAR-10 experiments we use a conditional reconstructor
which takes head-NN's parameters concatenated with the one-hot encoded class vector that determines
the class of the output image, enabling one reconstruction per class (in one-shot TL this reconstructs
the entire training set). This is sufficient to constitute a serious security breach.

**Reconstructor NN Architecture**     We have used a (modified) image generator architecture from
Gulrajani et al. (2017); Wu et al. (2020), see Appendix G. This approach is widely used in the
(generative) model inversion field (Fredrikson et al., 2015; Zhang et al., 2020; Yang et al., 2019;
Wang et al., 2021; Zhu et al., 2023; Liu et al., 2024; Ye et al., 2024) enabling high quality sampling
from the data generating distribution of the training data, see Section A. This architecture naturally
scales to higher-resolution input and reconstructions beyond the $32 \times 32$ setting used in prior work.

**Weight Extraction in the Black-box Attack**     In the black-box setting, head-NN's weights are not
public. To extract them we run the hard-label black-box weight extraction attack by Carlini et al.
(2025) that extracts hidden layers' weights (in the black-box experiments the head-NNs have one
hidden layer, see Table 1) which we feed into the reconstructor NN. The weights are reconstructed
only up to neuron permutations and rescalings (see Appendix I.4. This ambiguity causes the black-box
reconstruction to be less effective than its white-box counterpart, see Table 2.

We train the image classifiers on balanced sets i.e., $N = C \cdot M$, where $C$ is the number of classes.
For CIFAR-10 and MNIST, $C = 10$ with $M$ data-points per class. In CelebA-experiment (binary
face recognition task) each classifier is trained on $M$ images of the target identity and $9M$ images of
random others; identities vary across shadow models. We train $N_{shadow}^{train} = 2.56 \times 10^6$ training shadow
models and $N_{shadow}^{val} = 10^3/10^4$ validation shadow models for the CIFAR-10, MNIST/CelebA
experiments respectively. From training-shadow models we also calculate the mean $\bar{\theta}$ and the
covariance matrix $\mathrm{Cov}_\theta$ of the weights (flattened and concatenated across layers). We normalize the
input of the reconstructor NN as $\theta_{k,i} \rightarrow (\theta_{k,i} - \bar{\theta}_i)/\sqrt{[\mathrm{Cov}_\theta]_{i,i}}$ due to the possible variation in the
magnitude of the trained weights. In estimating $TPR$ and $FPR$ of the reconstructor NN we use the
Neyman-Person criterion, see Algorithm 4 in Appendix F.

**Reconstructor NN Loss Function**     The robustness and effectiveness of our attack (unseen in
prior work, see Appendix I.3) comes from shifting the focus to single-instance reconstruction: for
each class, the reconstructor NN discovers the data point most prone to reconstruction. To realize
this, we introduce the *softmin*-loss that automatically selects the most vulnerable instance among
$N$ candidates, also circumventing the data permutation ambiguity. Let $(\mathcal{D}_k, \theta_k) \in \mathcal{D}_{shadow}$ and
let $\mathcal{D}_k^c = \{Z_1^c, \ldots, Z_M^c\}$ be the subset of $\mathcal{D}_k$ consisting of images of the class $c \in \{0, \ldots, C-1\}$.
Let $\theta_k^c$ be the vector $\theta_k$ appended with the one-hot encoded class vector of the class $c$. Given a
reconstruction $R(\theta_k)$ we define the reconstruction loss of the conditional reconstructor (for CelebA
we always take $c = 1$, the class corresponding to one specific individual) via the following *softmin*
function.

$$loss_{rec}\left(\mathcal{D}_k^c, R(\theta_k^c)\right) = \frac{\sum_{i=1}^M \ell_i \exp\left(-\alpha \, \ell_i\right)}{\sum_{i=1}^M \exp\left(-\alpha \, \ell_i\right)}, \quad \alpha > 0, \tag{1}$$

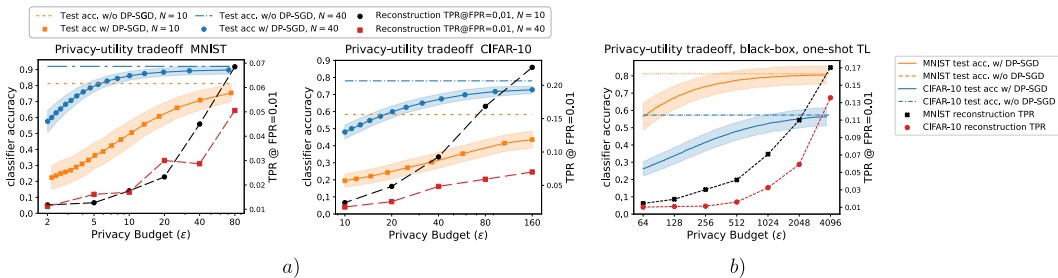

Figure 3: Privacy-utility tradeoff for the MNIST and CIFAR-10 experiments: a) white-box attack, b) black-box attack. All models suffer severe accuracy degradation even for relatively large values of $\epsilon$.

where $\ell_i \left( Z_i^c, R(\theta_k^c) \right)$ is the sum of the mean squared error (MSE), mean absolute error (MAE) and the LPIPS loss (Zhang et al., 2018) (also used in Balle et al. (2022)). In our experiments we have worked with $\alpha = 100$. The (soft)-minimum in this loss function is taken over the target images preventing the reconstructor from outputting aggregates of several targets (see Figure 19 in Section H for examples). For the reconstructor-NN training we use Adam optimizer (Kingma & Ba, 2015) (learning rate 0.0002 and batch size 32). Early stopping after about $10^6$ gradient steps prevents overfitting. This takes up to 72 hours to train on a GeForce RTX 3090 GPU. See Appendix I.5 for detailed attack timings which never exceed 80 hours (including shadow model generation on multiple cores). Note that a determined adversary can possess considerably more resources than this.

For a flowchart summarizing the reconstructor-NN training and attack, see Figure 13 and Figure 2 in Appendix G. Reconstruction examples and with $TPR$ values are presented in Table 2 and in Fig. 1. More reconstruction examples and further details of the experiments see Appendix H and Appendix I. Our attack is robust under a wide range of circumstances which is unseen in prior work: shadow model budget variations, varying classifier's training set size, reconstructor conditioning mismatch, varying classifier's training hyperparameters (SGD/Adam, weight initialization, underfitting/overfitting, data augmentation), and out-of-distribution mismatch between data priors for training- and validation-shadow models. Detailed ablation studies for the white-box attack are presented in Appendix I.3.

## 3.2 OUR RECONSTRUCTION ATTACK UNDER DP-SGD

DP-SGD adds noise to gradients during training to provably protect training data from reconstruction (Abadi et al., 2016). We study defense against the (realistic) weak adversary and show that, in the few-shot regime, effective protection requires very noisy DP-SGD, sharply degrading utility. Fig. 3 shows the privacy–utility tradeoff and our attack's effectiveness in the MNIST and CIFAR-10 experiments. The experimental setup was the same as in Section 3. We train with mini-batch DP-SGD (Gaussian noise; gradients clipped at $l_2$-norm $C$) to achieve $(\epsilon, \delta)$-DP with $\delta = N^{-1.1}$ as recommended in Ponomareva et al. (2023). We make use of the privacy amplification enabled by the mini-batch Poisson sampling with $q = (N-1)/N$. The calibrated $C$ values were 4.0 (MNIST) and 1.2 (CIFAR), see the details in Appendix J.

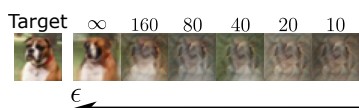

Figure 4: Reconstruction example for CIFAR-10 and $(\epsilon, \delta)$-DP models for training set size $N = 10$. Non-private training means $\epsilon = \infty$.

**The $TPR$-at-low-$FPR$ criterion** To find the value of $\epsilon$ which is sufficient to defend against our reconstruction attack, we look at $TPR$ at low $FPR$ (the "low" value of $FPR$ is somewhat arbitrary – here, we take 1%). As plots in Fig. 3 indicate, the $TPR$ drops when $\epsilon$ grows. Thus, to decide if our attack has been defended against, one can select a threshold $\gamma > 0$ and test the criterion $TPR \leq \gamma$ at $FPR = 0.01$. Smaller $\gamma$ means stricter defense criterion. In our experiments we have found that $\gamma = 0.03$ is sufficient to remove almost all of the details of the original data from the reconstructions. By interpolating data from Fig. 3a for white-box attacks this translates to a privacy budget of $\epsilon \approx 22.9$ and $\epsilon \approx 27.9$ for MNIST with $N = 10$ and $N = 40$ respectively. For CIFAR-10 we get $\epsilon \approx 12.6$ and $\epsilon \approx 19.3$ for $N = 10$ and $N = 40$ respectively. As

can be read off from the test accuracy plots in Fig. 3a, defending a training set of the size $N = 10$ against our attack causes model's accuracy to drop by over 30 p.p. both for MNIST and CIFAR-10. When $N = 40$ the accuracy drop is smaller, but still notable – slightly above 10 p.p. for MNIST and 18 p.p. for CIFAR-10. Fig. 4 illustrates how different values of $\epsilon$ affect a CIFAR-10 reconstruction for $N = 10$. One can see that at the threshold value where DP-SGD is deemed to have succeeded ($\epsilon$ between 10 and 20) the reconstruction becomes very blurry and for higher $\epsilon$ some features of the target are still being reconstructed indicating a privacy breach (this is especially evident when $\epsilon = 160$ – the reconstruction $TPR$ is still above 23%). For more white-box reconstruction examples under DP-SGD see Appendix J. For black-box attacks from Fig. 3b (one-shot TL) the threshold of $\gamma = 0.03$ is reached at $\epsilon \approx 280$ and $\epsilon \approx 950$ for MNIST and CIFAR-10 respectively. This results in accuracy drop of 9 p.p. and 8 p.p. for MNIST and CIFAR-10 respectively – the privacy-utility tradeoff is non-negligible, albeit considerably less pronounced than in the case of white-box attacks.

## 4 REVERSE HOMOMORPHIC ENCRYPTION

**Conventional Homomorphic Encryption (HE)** HE was proposed by Rivest et al. (1978) where the notion 'homomorphism' in the context of cryptography was first introduced. Gentry (2009) gave the first functioning fully homomorphic scheme, prompting a flurry of subsequent schemes. For readers new to HE (Ko, 2025; Cheon et al., 2017; Yagisawa, 2015; Fan & Vercauteren, 2012; Marcolla et al., 2022), here is a brief overview sufficient for understanding the main concepts in this paper. HE enables computation directly on encrypted data (which is called ciphertexts): encryption uses a public key (i.e. a key that is freely available without compromising the security of encrypted data) and decryption uses a private key that must be held by a trusted party because its release would compromise the security of the process.

The goal of HE is full data confidentiality – for example, when outsourcing computation to the cloud without revealing the sensitive data. This is achieved by sending encrypted data to the cloud, conducting all cloud computations under HE, and then retrieving the encrypted result and decrypting it with the private key within a trusted environment. Fig. 5(a) shows this process as applied to transfer-learned network predictions. To im-

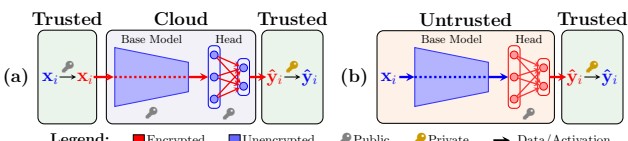

Figure 5: (a) Shows conventional HE usage in the context of performing TL neural network predictions. (b) Shows how RHE is implemented.

plement encrypted computations, HE preserves addition and multiplication in a homomorphic way i.e., the encryption of data $m$ to obtain ciphertext $c = E(m)$ satisfies $E(m_1 + m_2) = E(m_1) + E(m_2)$ and $E(m_1 \cdot m_2) = E(m_1) \cdot E(m_2)$. So, any circuit (i.e. sequence) of additions and multiplications can run on such ciphertexts. The foregoing description is highly simplified: in practice, software packages support a range of operations apart from addition and multiplication such as subtraction, rotation and an ability to carry out operations in vectorized form. However, practicality and efficiency remains a major challenge and an active research area. HE requires injecting noise into operations; without this it lacks semantic security. Noise grows with algorithm/circuit depth, i.e. with the number of homomorphic computations (most critically multiplications) performed in succession with outputs from one layer feeding into inputs of the next. After reaching a certain circuit depth very expensive 'bootstrapping' reset operations must be used to remove the effect of the noise. Thus inference in deep neural nets can be prohibitively expensive under HE without high-end hardware. Under HE, shallow neural nets have per-prediction times ranging from tenths of a second to several seconds on standard hardware Brutzkus et al. (2019) while older methods take $\sim 10^2$ s (Dowlin et al., 2016). Deeper/larger nets (e.g., ResNets) need several thousands of seconds per (single-threaded) prediction (Lee et al., 2022). Section C reviews more advanced accelerated approaches using high performance hardware.

### 4.1 DESCRIPTION OF REVERSE HOMOMORPHIC ENCRYPTION (RHE)

We assume partial fine-tuning which means the following **assumptions for the TL process**: i) the transfer learned head-NN is fully connected with at most three layers and ii) no base-model

finetuning is applied. One can produce highly accurate models operating within these constraints – few-shot TL using even just a single-layer (linear) head-NN without base-model finetuning can achieve competitive performance (Chen et al., 2019c; Tian et al., 2020).

RHE is most efficient when applied to only a few top layers, suggesting that, when training with sensitive data, practitioners should prefer partial fine-tuning to preserve per-prediction efficiency. Such techniques are well established and can match or closely approach full-model fine-tuning accuracy when the base model is sufficiently powerful, see Kornblith et al. (2019). Fine-tuning and encryption of more layers quickly increases the per-prediction latency of the model, supporting our choice to focus on shallow heads. Figure 6 collects selected results for HE schemes that encrypt inputs/activations but keep weights plaintext. Prior work (e.g., Aharoni et al. (2023)) shows that additional weight encryption increases latency by about 15–25% with careful implementations, a useful proxy for RHE scaling for deeper head-NNs or even full model fine-tunes.

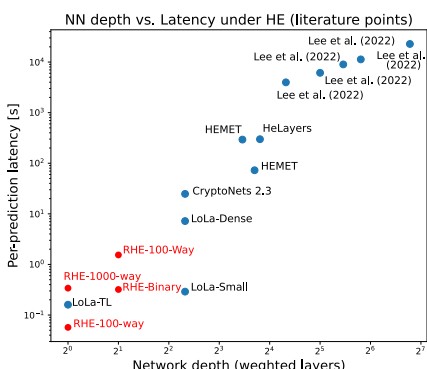

Figure 6: Per-prediction latency of HE encryption schemes vs. network depth defined as the number of the weighted layers. Data collected from the papers introducing the schemes: LoLa (Brutzkus et al., 2019), HeLayers (Aharoni et al., 2023), HEMET (Lou & Jiang, 2021), CryptoNets 2.3 (implemented in Brutzkus et al. (2019)), and Lee et al. (2022).

Our novel implementation of HE which we call 'Reverse Homomorphic Encryption' (RHE) keeps (inference query) input data *unencrypted* and instead encrypts weights and biases in the head-NN. Because only the head-NN depends on the TL data (the base model is frozen), encrypting these parameters blocks any reconstruction attack that inverts the weights back to the TL training data. Importantly, black-box attacks that rely on querying head-NN input–output pairs (such as that in Section 3) are also neutralized by RHE, since outputs are encrypted.

**RHE Workflow** TL network training is done in a trusted environment (without HE). Then, we homomorphically encrypt the weights and biases of the head-NN using the public key and release the encrypted model (and public key) to an untrusted environment where an adversary may intend to recover TL training data. The inference process of the part-encrypted network is shown in Figure 5(b). It proceeds as follows: unencrypted images pass through the unencrypted base model in the standard way. Once the (unencrypted) deep features enter the encrypted head-NN, all operations run under HE (this being facilitated by use of the public key), producing encrypted logits while never exposing head-NN parameters. As well as white-box attacks, this defeats *any* black-box attack for recovering the TL training data because an adversary is unable to query the classifier. Finally, a trusted party who operates within a safe environment decrypts the logits with the private key enabling readout of the unenciphered classifier logit outputs and decisions. Note that as the base model is not fine-tuned, observing its intermediate unencrypted layers reveals nothing about the TL training data.

### 4.2 COMPUTATIONAL EFFICIENCY AND ACCURACY OF RHE

The key reason for favourable RHE efficiency is that HE is applied only to a shallow head-NN ($\leq 3$ layers), avoiding costly bootstrapping. We also replace ReLU activations (expensive to implement in HE) with the square activations $f(x) = x^2$ which are cheap to implement in HE. As shown in Section D and Table 4, this preserves accuracy, helped by the shallow depth of the head-NN. A further speed-up is achieved by skipping the HE *argmax*-function to obtain hard-class decisions: encrypted logits are sent to a trusted party, who decrypts them and computes the classifier decisions in a trusted environment where HE is unnecessary.

**Implementational details:** For our proof-of-concept RHE implementation we use CKKS homomorphic encryption Cheon et al. (2017) and the TenSEAL library (Benaissa et al. (2021)). Furka et al. (2023) provides good guidelines on using the CKKS scheme with TenSEAL, in particular on how to select the CKKS parameters that affect the security, accuracy and computational efficiency of

| Benchmark | Head | | Mean Inference Time per Prediction | | |
|---|---|---|---|---|---|
| | Input Dim. | Size | Base Model | Head | |
| | | | | Unencrypted | Encrypted |
| MNIST | 256 | 10 | $0.5\,ms$ | $0.008\,ms$ | $8\,ms$ |
| CIFAR-10 | 1280 | 10 | $55\,ms$ | $0.01\,ms$ | $27\,ms$ |
| CelebA | 2048 | $4-1$ | $30\,ms$ | $0.02\,ms$ | $56\,ms$ |
| Linear | 2048 | 100 | – | $0.02\,ms$ | $57\,ms$ |
| Linear | 2048 | 1000 | – | $0.04\,ms$ | $0.34\,s$ |
| 1 HL (binary) | 2048 | $128-1$ | – | $0.03\,ms$ | $0.32\,s$ |
| 1 HL (16-way) | 2048 | $128-16$ | – | $0.04\,ms$ | $0.98\,s$ |
| 1 HL (100-way) | 2048 | $128-100$ | – | $0.07\,ms$ | $1.54\,s$ |

Table 3: Average time for a single prediction (no batching) averaged over 100 random input/weight choices. Hardware: Apple M4 Pro Laptop (8 cores). For linear heads the compute overhead due to RHE is comparable to forward propagation time through the base model.

the encryption scheme. We set the target security level at 128 bits, i.e., the only known attacks for breaking the homomorphic encryption require at least $2^{128}$ computer operations. Using the guidelines of Furka et al. (2023) we define the precision of our HE operations by specifying a 10 bit exponent and 23 bit fractional part and then set our encryption parameters accordingly. These parameters are chosen to ensure that the CKKS scheme meets the 128-bit security level and specified precision levels whilst at the same time achieving good efficiency. Further details are given in Section C.1 and summarized by a flow chart in Fig.7. Note that, as is evident from diagram Figure 5, we require encrypted-encrypted matrix-vector computations to allow encrypted weights to be combined with encrypted inputs, a situation that does not occur when using HE conventionally. We have extended TenSEAL's functionality to support such computations. Code and package installation instructions can be found here: `https://anonymous.4open.science/r/TenSEAL-65ED/README.md`

**Computational cost and accuracy:** For a given configuration of HE parameters the time complexity grows logarithmically with head-NN input dimension and as $\sim d\log_2 d$ with $d$ being the size of the intermediate layers (assuming all the dimensions are less than half of the polynomial modulus degree, see Appendix C.1). A more complete picture is presented in Section E. In Table 3 we provide (multithread) timings on a laptop: (a) for a range of network-head configurations including cases with one and two layers at and beyond what is typically sufficient for achieving competitive performance (Chen et al., 2019c; Tian et al., 2020), (b) for full implementations (i.e. unencrypted base model operations and network-head HE operations combined, as depicted in Figure 5(b)) of the RHE-secured versions of the transfer-learned networks analysed in Section 3. After switching to using the square activation function, which as already discussed has negligible effect (Section D), we found that using RHE caused **no degradation** of classifier performance in the MNIST, CIFAR-10 and CelebA experiments. Please refer to Appendix C for a discussion of possible speedups.

## 5 CONCLUSIONS, LIMITATIONS AND FUTURE RESEARCH

We have demonstrated new white-box and black-box training-data reconstruction attacks against few-shot transfer-learned classifiers. We have extended these attack capabilities to a realistic threat model and shown that DP-SGD cannot defend against them without severely damaging classifier utility. The compelling evidence we provide should act as a serious warning to both security advisors and practitioners. Given the growth in TL applications, including use of sensitive training data, this represents a serious gap in state-of-the-art defense capabilities. Our proposed solution is to use a novel homomorphic encryption-based defense, RHE, which is fully secure against reconstruction, MIA and property-inference attacks and is also surprisingly computationally efficient when applied to TL applications. Our laptop-only timings, using minor extensions of an open-source HE library, show that RHE is fully practical for inference without high-performance hardware. This contrasts starkly with most of the conventional HE usages in machine learning. We discuss limitations of our work and scope for future research in Appendix B.

ETHICS STATEMENT

We have not conducted research involving human subjects or participants. We only work with open source published datasets (CIFAR, MNIST, EMNIST, CelebA) and cite them accordingly in the paper. All of these datasets are available for non-commercial research purposes. For our reconstructor NN architecture we have used and modified the residual generator from Gulrajani et al. (2017); Wu et al. (2020) available under the MIT license (can be viewed on the respective project GitHub pages referenced in the papers). The potential harmful consequences of this work concern applications of our work where adversaries train their own reconstructor NNs to attack private datasets. However, none of our trained models can be directly applied to reconstruction of private datasets since our models were trained on public datasets and thus can reconstruct data from these open datasets only. What is more, we have proposed an encryption scheme that can prevent any data reconstruction attack.

REPRODUCIBILITY STATEMENT

We describe the technical details of all the experiments in Appendices C.1, G, I and J. Furthermore, we provide code and its documentation that allows one to reproduce all the experiments from the paper (requires access to a GPU and several CPUs) under the links provided at the beginning of Section I and Section C.1. The repositories include files containing hyper-parameter configuration for each experiment. The documentation specifies all the commands that are required to re-run the experiments. All the data we have used is either public or can be reproduced using the provided code. We have also published some of our trained reconstructor NN models together with test examples.

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

# A COMPARISON WITH PRIOR WORK

## A.1 PRIOR WORK ON TRAINING DATA RECONSTRUCTION

We study the problem of (partially) inverting the mapping between the training set and trained model's weights. The work (Balle et al., 2022) studies a similar problem to the one considered here by training a reconstructor NN. However, as explained above and specified in Appendix I.3, our attack works under a more realistic threat model and under much more general conditions. Our attack relies on a reconstructor NN giving reconstructions that are semantically close to the original images as opposed to the recently proposed gradient-flow based attacks (Haim et al., 2022; Buzaglo et al., 2023; Loo et al., 2024) which reconstruct the training data numerically as solutions to a certain algebraic minimization problem. Our attack is more flexible than the gradient-flow based methods which only work when the attacked model has been trained in a very specific way. For instance, they require long training with very low learning rates until good (approximate) convergence to a critical/KKT point (we do not need any of these assumptions). As such, the gradient-flow based attacks will be disrupted even by small amounts of noise in DP-SGD. The importance of the true- and false-positive attack success rates and ROC curves to evaluating deep learning privacy has been emphasized in the context of MIA (Carlini et al., 2022; Kulynych et al., 2024), however it has not been considered in data reconstruction attacks, especially in realistic threat models such as the weak adversary model. A hypothesis testing approach (different from ours) has been recently used to derive tighter reconstruction robustness bounds in the informed adversary model (Kaissis et al., 2023).

In this work we have used an image generator architecture as a reconstructor-NN. This approach has been widely adapted in the field of (generative) model inversion (GMI) which leverage GANs (Fredrikson et al., 2015; Zhang et al., 2020) or diffusion models (Liu et al., 2024) to reconstruct information about the training set of the target classifier. The goal of GMI is to recover representative samples from the generating distribution of the classifier's training data (general representative 'prototypes' of data samples associated with given labels) (Yang et al., 2019; Wang et al., 2021; Zhu et al., 2023). This is different from our goal which is to obtain *exact instances* from the training set. For instance, when attacking a face recognition model, GMI would generate fake samples of the face of the target person (Wang et al., 2021) (potentially enabling the attacker to fake that person's identity), whereas our method would output faithful reconstructions of the actual examples that have been used the attacked model training. This is a crucial difference – the notion of a 'positive reconstruction' would not provide useful information in the context of GMI, since it's goal is to obtain data 'prototypes' rather than exact data points. Thus, the reconstruction TPR, FPR and the related ROC curves which underpin the evaluation of exact/faithful reconstruction studied in this work would not provide useful information for evaluating GMI. GMI provides an alternative methodology allowing one to sample representative images of high quality (i.e. realistically looking and not blurred) both in the white-box and black-box (including hard-label black-box) settings (Liu et al., 2024; Ye et al., 2024) which can also reveal potentially useful characteristics of the training data of concern to privacy, but not near copies of training data instances as provided by our method. Note that our RHE scheme defends against any GMI attack on TL training data by encrypting all the information about the training data.

Salem et al. (2020) study exact logit-based black-box data reconstruction attacks in an online learning setup, where the target/attacked model's weights are being continuously updated using small amounts of new data. This setting is similar to TL in the sense that it considers effective training on small amounts of data. Salem et al. (2020) also uses the shadow model method to train a reconstructor-NN which takes as its input the output logits of the target model when queried on a fixed input data sample. This is different from our black-box setting which uses only hard-label decisions of the target model rather than its output logits, thus assumes that the adversary has access to much less information. Salem et al. (2020) do not address the possibility of false positive reconstructions and do not test their attack against DP-SGD. To the best of our knowledge, our work is the first to realize the more difficult hard-label black-box exact training data reconstruction attack which produces high quality faithful reconstructions and is robust against DP-SGD.

## A.2 PRIOR WORK ON HOMOMORPHIC ENCRYPTION

As we have explained in Section 4.2 the computational efficiency of our RHE scheme relies on the fact that RHE applies the minimal amount of encryption that is sufficient to fully defend the model

against training data reconstruction (and other related) attacks in TL. In particular, since our goal is not to protect the input queries during inference (this data is completely separate from the TL training data and assumed to not be private in RHE), in RHE we do not encrypt the base model outputs or its intermediate activations which saves a great amount of compute. Note that if one wants to keep user's input queries private while outsourcing computations to the cloud/server, one should also encrypt all the intermediate activations and outputs of the base model since leaking any of this information leaves the model susceptible to reconstruction using inversion from deep activations (Mahendran & Vedaldi, 2015; Dosovitskiy & Brox, 2016).

Whilst conventional HE has been used to homomorphically encrypt some or all of model's weights when running on encrypted data (the so-called oblivious or partially oblivious inference (Orlandi et al., 2007; Rizomiliotis et al., 2022; Kim & Guyot, 2023)), such an approach in traditional settings is prohibitively expensive without access to high-end compute hardware, unless some simplified neural networks are used (Zhang et al., 2024). They often also need to use multi-party computation Liu et al. (2017); Rathee et al. (2020); Srinivasan et al. (2019); Juvekar et al. (2018).

HE methods in TL often focus on the privacy challenges arising when a server stores a model and lets the clients query results for their private input data and enables clients to perform transfer-learning on their private training datasets (Jin et al., 2020; Lee et al., 2023). Within this framework, schemes encrypting only the transfer-learned weights of the final classification layer and the base-model outputs (i.e. the deep features) have been proposed both for TL training and inference by Lee et al. (2023). More precisely, a user/client who wants to protect both the TL training data and their query data at inference computes the base-model outputs locally in their trusted environment in an unencrypted way, encrypts them and sends them to the (untrusted) server/cloud who performs both the TL training and inference on the ciphertexts, sending the encrypted output logits and the encrypted weights of the final classification layer to the client for validation. We re-emphasize that this is different from our framework where the query inputs at inference are not private and the goal is the protection of the TL training data only, see Figure 5. This means that:

1. The set of use cases of RHE is disjoint from the set of use cases of HETAL by Lee et al. (2023) since we are not keeping the inputs at inference private. Note that this often occurs in practice e.g. in public social media content moderation or when the TL training uses licensed data (but the model is queried on public data).

2. In RHE TL training is done without encryption within a trusted environment requiring the trusted user to have expertise in TL. Lee et al. (2023) proposes a HE scheme for TL training in an untrusted environment instead which allows the client to outsource the training to an expert service provider. However, the service provider must share the encrypted intermediate TL weights/logits with the client who calculates the validation loss to decide when to stop the training. This requires the client to have some limited level of expertise in TL. The service provider also has limited knowledge about the underlying problem since the data they receive is encrypted, thus they may not be fully capable of applying their expertise to TL training.

3. The emphasis of Lee et al. (2023) is to protect training data during training which is done in an untrusted environment, but our emphasis is to defend against training data reconstruction attacks at inference.

4. In RHE at inference the deep features remain unencrypted, enabling us much more flexible use cases since any party can carry out predictions, while in the setting of Lee et al. (2023) the query input data remains private, thus only the client can query the model.

5. The setting of Lee et al. (2023) requires the client to extract the deep features locally which burdens the client with doing potentially expensive forward propagation through the complex base model e.g., a vision transformer which requires high-end hardware to run (Dehghani et al., 2023; Oquab et al., 2024).

6. In RHE the entire TL model (unencrypted base-model plus the encrypted head-NN) is deployed within an untrusted environment (e.g. deployed for use by multiple teams at different sites within an organisation or kept on an untrusted server).

7. By focusing purely on the problem of inference, we are able to put fewer restrictions on the head-NN architecture (at most three fully connected layers with square activations ) when

compared to Lee et al. (2023) which considers fine-tuning of only one final classification (*softmax*) layer.

This approach allows us to identify the minimum amount of encryption required to implement a fully effective and targeted defense against TL training data reconstruction during inference. This efficient minimalist approach permits prediction times that are comparable to the forward propagation through the (unencrypted) base model (we avoid repetitive bootstrapping and expensive activation functions that are entailed by applying HE throughout the whole network), see Section C for more details.

## B  LIMITATIONS AND FUTURE WORK

This work has focused on attacks and defense in transfer-learned image classification. This is an important and widespread area of application in its own right and one in which the availability of large and diverse image data-sets for construction of base models facilitates use of transfer-learning across a range of important applications. This leaves scope for future research on the use of our attack and defense techniques in other domains such as text and speech.

One limitation of our data reconstruction attacks is that they become less effective as the size of the attacked model's smallest per-class training set increases. Our attacks have been shown to be effective in cases where the minority class has 1-4 training images. When the class from which we recover a reconstruction becomes larger than this the reconstruction $FPR$ increases making our attacks unreliable. It is not clear whether this threshold reflects a fundamental limitation for any attack carried out under the weak adversary threat-model. Further attack enhancements will allow us to investigate whether it can be shown that larger per-class cases can be attacked without DP defense. Meanwhile we believe that our results as given, and the inability of DP-SGD to effectively defend against our attacks, already provide a compelling reason for practitioners of few-shot TL to look to RHE as a fully effective defense.

Another limitation of our current attacks concerns the architecture of the reconstructor NN and the related amount of compute and memory required to train it. Recall that the input of the reconstructor NN consists of the concatenated and flattened weights and biases of the image classifier layers that have been trained during the transfer learning. This is typically only one or two final layers, however as explained in Table 6 the resulting input dimensions $D_\Theta$ can easily be of the order of $10^4$. This means that most of the parameters of the reconstructor NN are contained in its first convolutional layer. The exact number of parameters in this layer is equal to $4S(16D_\Theta + 1)$ where $S$ is the internal size of the reconstructor NN (equal to $256$ or $512$). Thus, the first convolutional layer contains $\sim 10^8$ parameters. This makes the training relatively inefficient. In future work we will explore more efficient ways of processing reconstructor NN input. For instance, instead of concatenating all the weights of the trainable layers one could take inspiration from Deep Sets (Zaheer et al., 2017) and process each neuron separately by a sequence of convolutions and sum the result over the neurons. Similar approach has been used to construct NNs used for property inference attacks (Ganju et al., 2018). By processing the neurons in a permutation-invariant way, this could also improve the effectiveness of our black-box attack by addressing the problem of neuron permutation ambiguity in the weight recovery via the hard-label black-box weight extraction.

Another limitation of our work is that we have not proposed any defense for defending the (large) training dataset of the base model. In practice, the base models are generally trained on public data in which case no defenses are required. However, even if the base model is trained using a sufficiently large private dataset, DP-SGD may readily provide sufficient defense (possibly subject to prior pretraining on a related public dataset), see for instance Sander et al. (2023); Ganesh et al. (2023).

Whilst the per prediction times of RHE-defended networks have been shown in section 4.2 to be practical there is still scope to improve speed both on low-spec and high-spec hardware implementations. Section C provides details. Such improvements only seem likely to be of potential value for the more challenging, and less commonly used, instances of 2 and 3 layer fully connected network heads.

At first sight the need to deal with encrypted classifier output under RHE might appear to be an unnecessary encumbrance. However this should not be viewed as a limitation since our demonstration of black-box attacks shows encryption of classifier output to be a necessity. One unavoidable "limitation" is that the trusted party must not reveal the class-decision output to an untrusted party,

e.g by allowing the untrusted party to monitor their follow-on action. However this is a fundamental and unavoidable requirement given that DP measures are unable to block black box attacks without unacceptable damage to classifier utility

# C   MORE ON HOMOMORPHIC ENCRYPTION

HE allows operations on encrypted data. The most dominant HE schemes are CKKS encryption by Cheon et al. (2017), BGV encryption by Yagisawa (2015) and BFV encryption by Fan & Vercauteren (2012). BGV and BFV encryption are quite similar and so are often mentioned somewhat interchangeably. The reason these schemes dominate is because they are fast, allow one to pack multiple numbers (vectors) into a cyphertext, can carry out encryption on fairly deep networks of operations without the need for costly 'bootstrapping' operations and are widely supported by software libraries. In this paper we focus on the CKKS scheme which supports the following operations on real numbers expressed in floating point (rather than integer) form: Addition, Multiplication, Subtraction and Rotation. We focus on HE performance in inference and not during neural net training which we assume is carried out in a trusted environment using unencrypted operations. Gilad-Bachrach et al. (2016) were one of the first to use a neural network to perform inference on HE data. Later this was extended by Brutzkus et al. (2019) leading to dramatic speedups in the time for inference and reduction in memory footprint by better handling vectorization. The CKKS with TENSeal (Benaissa et al., 2021) draws upon their Stacked Vector–Row Major scheme for matrix-vector computations.

HE is widely regarded as too slow, even for inference. Most of this slowdown comes after reaching maximum multiplicative depth, in neural networks this manifests itself as network depth, at which point one needs to 'bootstrap' which is extremely computationally expensive, see Kang et al. (2020) for some timings which in some cases were found to exceed 100 seconds. The original bootstrapping algorithm for CKKS can be found in Cheon et al. (2018). A newer variant which is implemented in Microsoft's SEAL library (SEAL) is the work of Chen et al. (2019a). Gong et al. (2024) provide a survey detailing work to speed up HM addition, HM multiplication, number-theoretic transform (used to perform HM operations) and bootstrapping, including the use of hardware acceleration on GPUs for example. However, much of this work remains in progress and isn't easily implementable.

By staying within the confines of a three-layer fully connected network with ReLU activations replaced by square activation functions we bypass the need to invoke bootstrapping in our HE implementations. This approach is similar to the one adopted in leveled HE schemes. Whilst there are ways to speed up inference by taking advantage of parallel computing environments and specialized hardware our aim is to demonstrate practicality for the user who may not have access to these facilities.

Gong et al. (2024) provide a survey on accelerating HE, we will provide a short discussion now. Without any hardware acceleration there are things one could implement in order to speed up computation. One could look at the configuration of the HE scheme Cheon et al. (2024). One could also look at implementing faster versions of computations used in neural networks such as implementing faster encrypted-plain (Bae et al., 2024) and encrypted-encrypted (Park, 2025) matrix multiplications (these works have been successful in speeding up times when matrices are very large).

Recently lots of work has also been done on accelerating HE operations on specialized hardware. For example lots of work has been done on making HE work on GPUs (Choi et al., 2024; Fan et al., 2023; Agulló-Domingo et al., 2025; Shivdikar et al., 2023; Jung et al., 2021; Fan et al., 2025). Work has also been done on running HE on FPGAs, which have been able to achieve massive speedups, such as the work of Samardzic et al. (2021) and Xu et al. (2025). Some have even tried accelerating HE using ASICs (Samardzic et al., 2022; Kim et al., 2023; Aikata et al., 2023).

If one wants to use deeper networks there are also recent works which look to make this more feasible. The work of Yan et al. (2025) propose a novel rescaling operation that saves on the depth that bootstrapping requires and speeds it up. Jung et al. (2021) are able to dramatically speed up bootstrapping by multiple orders of magnitude (which is critical should you want deeper networks) using GPUs. Rovida & Leporati (2024) studies HE-friendly polynomial activation functions, see also Njungle & Kinsy (2026) for comparison of different activation functions optimized for HE. Some recent advances in bootstrapping include the development of programmable bootstrapping for the

Fast Fully Homomorphic Encryption over the Torus (TFHE) (Chillotti et al., 2021) and functional bootstrapping for CKKS (Alexandru et al., 2025).

### C.1 CONFIGURING HOMOMORPHIC ENCRYPTION

To encrypt the head of our network we use CKKS encryption and the TenSEAL library (Benaissa et al. (2021)). There are two parameters which need to be set to produce a configuration for HE. These parameters effect the security, performance, and computational depth of our encryption scheme. To reproduce HE experiments presented in this paper, please use the following repository:

- Reverse Homomorphic Encryption `https://anonymous.4open.science/r/RHE-A7CF`.

The first is the polynomial modulus degree ($N_D$) which defines the degree of the polynomial ring used in the encryption scheme, it must be some power of 2. The larger the polynomial degree the higher security level provided. For a larger polynomial modulus degree we also get a larger slot capacity i.e., the number of values that can be packed into a single ciphertext and which equals $N_D/2$. By packing our ciphertexts we also enable Single Instruction, Multiple Data (SIMD) operations. The higher the degree, $N_D$, the deeper the computational circuits we can in theory use without bootstrapping [1].

The second parameter is the coefficient modulus bit sizes which is a list of integers, $Q = [Q_S, Q_M, ..., Q_S]$ where there are $\mathcal{M}$ copies of $Q_M$. $\mathcal{M}$ is the depth of the multiplicative circuit, ie. it tells us how many multiplicative operations can be performed in sequence before bootstrapping is required. In TenSEAL we must have $Q_M \geq 20, Q_S \geq Q_M + 10, Q_M, Q_S \leq 60$. The values of $Q_S$ and $Q_M$ affect the precision maintained during computations ((Furka et al., 2023)). Specifically the bit-precision of the exponent, $\eta_I$, and the bit-precision of the significand, $\eta_D$, of each number stored in the ciphertext.

The total sum of all bit sizes, $Q_{max} = 2Q_S + \mathcal{M}Q_M$ (see Figure 7), must remain below a security bound for the chosen polynomial degree, $N_D$. The security level, ($\lambda$), is measured in bits and a commonly used value is 128 bits meaning that with known attack capabilities an attacker would need to perform at least $2^{128}$ computational operations to break the encryption. In practice this is computationally infeasible.

Furka et al. (2023) provides good guidelines on using the BFV and CKKS schemes with TenSEAL. In particular, one should have as short a list of coefficient modulus bits as possible based on the number of homomorphic operations that one will perform. The same goes for $N_D$ which also determines slot capacity (equals $N_D/2$). These choices are also subject to ensuring that the desired security level ($\lambda$) is achieved. We chose our configuration for HE using the recipe in Section E in Furka et al. (2023) as detailed in Figure 7. For further reading on parameter choices please refer to Chase et al. (2017); Albrecht et al. (2021).

Once the HE has been configured, to perform RHE all one must do is encrypt the weight matrices and bias vectors in the head using the public key. If the input to a layer is larger than $N_D/2$ or if $\lfloor N_D/2k \rfloor$, where k is the input size to a layer, is greater than 1 for any layer then one must perform chunking, this is made clear in Section E. Chunks can be computed in parallel to speed up the forwards pass. Otherwise one simply encrypts bias vectors and flattens matrices row-wise before encrypting the resulting vector. For details on how to perform a forward pass through the encrypted head please refer to Section E.

## D REPLACING THE RELU ACTIVATION FUNCTION

For all the benchmarks used in the attack we investigated what happens to performance when we replaced the ReLU activation function with the square activation function. To make a fair comparison we conducted an iterative grid search over hyperparameters for both activation functions for each benchmark and present in Table 4 the best results from the grid search. As shown in Table 4 the test

---

[1]In TenSEAL there is a hard limit of circuit depth set to 7 without bootstrapping (hence we confine ourselves to a maximun of 3 layers, which is more than we need). Although in theory we can have arbitrary circuit depth by changing $N_D$ in practice this is infeasible. With a larger degree comes increasingly longer computation time as $N_D$ must be some power of 2 and a larger $N_D$ leads to longer computation times for HM operations.

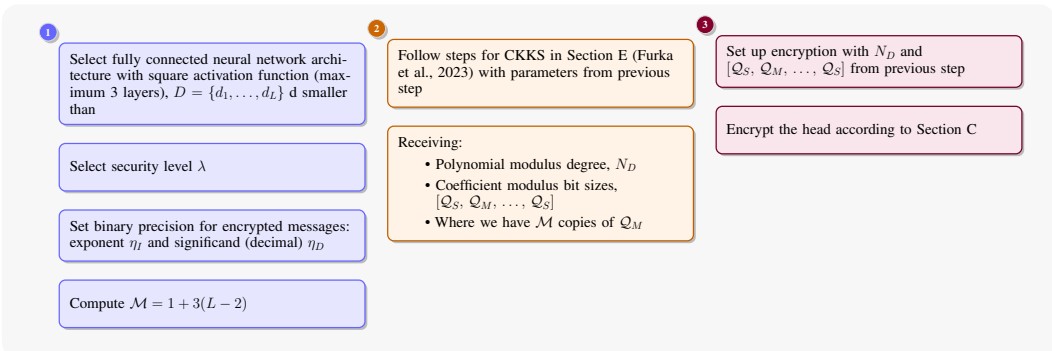

Figure 7: Figure detailing steps for configuring RHE with TenSEAL.

| Experiment | $N = 10$ | | $N = 40$ | |
| | ReLU | Square | ReLU | Square |
|---|---|---|---|---|
| MNIST | $83.5 \pm 4.1\%$ | $81.4 \pm 4.1\%$ | $91.6 \pm 1.8\%$ | $88.5 \pm 7.3\%$ |
| CIFAR-10 | $90.7 \pm 3.2\%$ | $91.1 \pm 2.7\%$ | $97.0 \pm 1.3\%$ | $97.2 \pm 0.7\%$ |
| CelebA | $92.3 \pm 3.1\%$ | $92.1 \pm 2.9\%$ | $94.2 \pm 3.1\%$ | $94.2 \pm 2.9\%$ |

Table 4: Accuracy on validation set averaged over 100 models each trained on a random sample (size $N = 10$ or $N = 40$) of the training data. Shown are best results after a systematic grid search over hyperparameter values.

accuracies achieved by both are very similar so that swapping the ReLU with the square activation function in the TL network-head appears to have negligible (positive or negative) impact on classifier accuracy.

Briefly commenting on optimal hyperparameters: networks trained with the square activation function performed better when using smaller learning rates and a larger number of epochs than the optimal hyperparameters found for ReLU. When training on benchmarks with $N = 40$ we kept the optimal hyperparamters found for $N = 10$ and only did a search over the number of epochs. For $N = 40$ we found both activation functions needed more epochs with square requiring a larger number of epochs than needed for ReLU most of the time.

# E    RHE INFERENCE TIME

One can completely and uniquely describe a fully connected neural network architecture by a sequence of dimensions $D = \{d_0, d_1, ..., d_L\}$ where $L$ is the number of layers (i.e. $L - 1$ is the number of hidden layers) and $d_l$, $l = 1, \ldots, L$ is the number of neurons in $l$-th layer. The first element of $D$, $d_0$ describes the dimension of the input to the first layer (i.e. the number of deep features) encountered in the head, the last element $d_L$ describes the output dimension of the network. Under this notation, each neuron in the $l$-th layer, $l = 1, \ldots, L$, has $d_{l-1}$ weights.

The operations we need consider in order to build a formula for timings are as follows:

1. Encrypted-matrix-plain-vector multiplication (supported in TENSeal)

2. Encrypted-matrix-encrypted-vector multiplication (not supported in TENSeal, but required for RHE, find our implementation here: `https://anonymous.4open.science/r/TenSEAL-65ED/README.md`)

3. Adding of bias term to encrypted vector (supported in TENSeal)

4. Repacking of chunked computation (supported in TENSeal)

5. Square activation function (supported in TENSeal)

Note that we ignore times for initializing vectors as this remains fairly constant and is independent of network architecture. An encrypted-matrix-plain-vector multiplication is computed using algorithm 2 and following the steps of the algorithm one can write the following formula for its computation time:

$$\lceil r/\lfloor N_D/2k \rfloor \rceil [t_{PM} + (\log_2(k) - 1)(t_{PA} + t_R)] \tag{2}$$

We use the fact that $n_c = \lceil r/\lfloor N_D/2k \rfloor \rceil$. Where $r$ is the number of rows/neurons in the layer, k is the number of columns or size of the input for the layer, $t_R$ is the time for a HM rotation and $t_{PM}$ and $t_{PA}$ are the times for a encrypted-plain HM multiplication and addition respectively.

Similarly an encrypted-matrix-encrypted-vector multiplication has equation:

$$\lceil r/\lfloor N_D/2k \rfloor \rceil [t_{EM} + (\log_2(k) - 1)(t_{EA} + t_R)] \tag{3}$$

Where $t_{EM}$ and $t_{EA}$ are the times for an encrypted-plain HM multiplication and addition respectively. Adding the bias term to an encrypted vector takes $\lceil r/\lfloor N_D/2k \rfloor \rceil \cdot t_{EA}$. Observing algorithm 3 one can see the time for repacking of a chunked computation is:

$$\lceil r/\lfloor N_D/2k \rfloor \rceil \cdot [t_{EM} + t_{EA} + t_R] \tag{4}$$

Finally the square activation function takes $t_{EM}$. Noting that the input dimension to a layer is padded to the next power of two we will write our input dimensions as $\hat{d}_i = 2^{\lceil log_2(d_i) \rceil}$ for parts of the equation that decide the chunking. Using this and the above equations we can provide the following formula:

$$C(i) = \begin{cases} \lceil d_{i+1}/\lfloor \frac{N_D}{2\hat{d}_i} \rfloor \rceil [t_{PM} + (\log_2(\hat{d}_i) - 1)(t_{PA} + t_R) + t_{EA}] & \text{for} \quad i = 0 \\ t_{EM} + \lceil d_{i+1}/\lfloor \frac{N_D}{2\hat{d}_i} \rfloor \rceil [2t_{EM} + (\log_2(\hat{d}_i))(t_{EA} + t_R) + t_{EA}] & \text{for} \quad L - 1 \geq i \geq 1 \end{cases} \tag{5}$$

Where the computational cost is given by $\sum_{i=0}^{L-1} C(i)$. One can see that the exact time complexity of RHE is complicated, however one can derive a simple practical upper bound (mentioned in Section 4.2). Assuming that $N_D - 4d_0 > 1, N_D - 4d_{\max} > 1$ we observe that:

$$\lceil d_{i+1}/\lfloor \frac{N_D}{2\hat{d}_i} \rfloor \rceil < 2 \cdot d_{i+1} \tag{6}$$

Also noting that $t_{PM} < t_{EM}, t_{PA} < t_{EA}$ we can write:

$$C(i) < \begin{cases} d_1[t_{EM} + \log_2(d_0)(t_{EA} + t_R) + t_{EA}] & \text{for} \quad i = 0 \\ t_{EM} + d_{i+1}[2t_{EM} + \log_2(2d_i)(t_{EA} + t_R) + t_{EA}] & \text{for} \quad L - 1 \geq i \geq 1 \end{cases} \tag{7}$$

Setting $d_{\max} = \max_{i=1,...,L} d_i$:

$$C(i) < \begin{cases} d_1[t_{EM} + \log_2(d_0)(t_{EA} + t_R) + t_{EA}] & \text{for} \quad i = 0 \\ t_{EM} + d_{max}[2t_{EM} + \log_2(2d_{max})(t_{EA} + t_R) + t_{EA}] & \text{for} \quad L - 1 \geq i \geq 1 \end{cases} \tag{8}$$

So the computation time of the first layer grows at most logarithmically with the input dimension, $d_0$, and linearly with its output dimension. The other layers grow at most at a rate of $d_{\max} \log(d_{\max})$. Usually the size of the hidden layers are larger than the output dimension and are set to be the same size. In which case we can say that the computation time of the first layer grows logarithmically with input dimension, linearly with hidden layer dimension and that the other layers grow at most with $d \log_2(d)$ where d is the size of the hidden layers.

One thing to note is that if one was to exceed in the input dimension to any layer the maximum slot capacity, $N_D/2$, extra steps are required. In the above analysis and in Algorithms 2 and 3 we assume that this isn't the case. This assumption isn't constraining as it's unlikely to crop up in TL. Even in our tests for networks at the extreme end of what you might expect to use in TL, all of the layers input dimensions fit well within the bounds of the maximum number of elements according to polynomial modulus degree. Even if this assumption is violated it is quite easy to fix: one would either have to increase the polynomial modulus degree or perform another level of chunking.

# F MEMBERSHIP INFERENCE SECURITY GAME, RECONSTRUCTION $FPR$, $TPR$ AND ROC CURVES

The Lemma below motivates the need for considering both $TPR$ and $FPR$ when evaluating data reconstruction attacks.

---

**Algorithm 2:** Matrix-Vector computation

---

**Input:** Vector $v$, polynomial modulus degree $N_D$, Matrix $M$ with $r$ rows and $k$ columns, broken into $n_c = \lceil r/\lfloor \frac{N_D}{2k} \rfloor \rceil$ chunks where each chunk contains $\lfloor N_D/2k \rfloor$ rows, $k$ number of columns and $d$ such that $k = 2^d$

1 Create vector $v_{stacked}$ with $\lfloor N_D/2k \rfloor$ copies of $v$ stacked.

2 **for** $i = 1$ **to** $n_c$ **do**

3    Fetch $m_i$, the encrypted vector containing the rows for the i-th chunk.

4    Compute $u = m_i \odot v_{stacked}$ (element-wise multiplication)

5    Initialize $result_i = u$

6    **for** $j = 1$ **to** $d - 1$ **do**

7       $result_i = result_i + \text{rotate}(u, 2^j)$

8    **end**

9 **end**

10 **return** *Concatenate all $result_i$ for $i = 1, \ldots, n_c$*

---

**Algorithm 3:** Vector Packing

---

**Input:** List of vectors $\{v_1, v_2, \ldots, v_\ell\}$, vector size $n$, number of vectors $\ell$

1 **Initialize:** Create packed vector $p$ of size $N_D/2$ with all zeros

2 Create mask $mask$ of size $N_D/2$ with ones in first $n$ positions, zeros elsewhere

3 Compute $output\_size = n \times \ell$

4 Compute $mask\_shift = output\_size - n$

5 **for** $i = 1$ **to** $\ell$ **do**

6    Compute $masked\_vector = v_i \odot mask$ (element-wise multiplication)

7    Update $p = p + masked\_vector$

8    Rotate $mask$ by $mask\_shift$ positions: $mask = \text{rotate}(mask, mask\_shift)$

9 **end**

10 **return** *Packed vector $p$*

---

**Lemma 1.** Let $\ell : \mathcal{Z} \to [0,1]$ be a normalised error function and $N$ be the size of the training dataset. Assume there exists $Z_0 \in \mathcal{Z}$ be such that $\kappa_\tau(Z_0) := \mathbb{P}_{Z \sim \pi}[\ell(Z, Z_0) \leq \tau]$ satisfies $0 < \kappa_\tau(Z_0) < 1$ for every error threshold $\tau \in ]0, 1[$. Define the *baseline attack* $\mathcal{R}_0$ as $\mathcal{R}_0(\theta) = Z_0$ for any input weights $\theta$. The cumulative $TPR$ and $FPR$ for this attack have the following pointwise limits for every $\tau \in ]0, 1[$.

$$TPR_{cum}(\tau) \xrightarrow{N \to \infty} 1, \quad FPR_{cum}(\tau) \xrightarrow{N \to \infty} 1.$$

*Proof.* Denote $\mathcal{D}_N = \{Z_1, \ldots, Z_N\}$ as a training dataset. We immediately find that

$$TPR_{cum}(\tau) = \mathcal{P}_{\ell_{\min} \sim \rho_0}[\ell_{\min}(\mathcal{D}_N, \mathcal{R}_0(\theta)) < \tau] = 1 - \mathcal{P}_{\ell_{\min} \sim \rho_0}\left[\min_{Z \in \mathcal{D}_N} \ell(Z, \mathcal{R}_0(\theta)) \geq \tau\right]$$

$$= 1 - \mathcal{P}_{\mathcal{D}_N \sim \pi_N}[\ell(Z_i, Z_0) \geq \tau, \, i = 1, \ldots, N] = 1 - \prod_{i=1}^{N} \mathcal{P}_{Z_i \sim \pi}[\ell(Z_i, Z_0) \geq \tau]$$

$$= 1 - (1 - \kappa_\tau(Z_0))^N.$$

Thus, by the assumption that $0 < \kappa_\tau(Z_0) < 1$, when the amount of training data is large we have $TPR_{cum}(\tau) \xrightarrow{N \to \infty} 1$. In other words, the baseline attack can achieve great reconstruction $TPR$. However, it is not retrieving any information about the actual training set since it is merely returning an arbitrary member of $\mathcal{Z}$ – using the same arguments we can show that its cumulative $FPR$ tends to 1 as well. $\qquad\square$

The $TPR$ and $FPR$ can be also interpreted in terms of a security game which proceeds between a challenger and an adversary.

1) The challenger samples a training dataset $D_n = (Z_1, \ldots, Z_N) \sim \pi_N$ and applies the training mechanism $D_n \xrightarrow{M} \theta$.

2) The challenger randomly chooses $b \in \{0, 1\}$, and if $b = 0$, samples a new dataset $D'_n \sim \pi_N$ such that $D_n \cap D'_n = \emptyset$ and applies the training mechanism $D'_n \xrightarrow{M} \theta'$. Otherwise, the challenger selects $\theta' = \theta$.

3) The challenger sends $D_n$ and $\theta'$ to the adversary.

4) The adversary gets query access to the distribution $\pi$ and the mechanism $M$ and outputs a bit $\hat{b}$ given the knowledge of $D_n$ and $\theta'$.

5) Output 1 if $\hat{b} = b$, and 0 otherwise.

Note that if the training mechanism $M$ is deterministic then the above game is trivial – the adversary can simply compute $M(D_n)$ and compare it with $\theta'$. Otherwise, the adversary can apply the standard MIA to some $z_0 \in D_n$ to decide whether $z_0$ is a member of the training set that corresponds to $\theta'$. In this sense, a reconstruction attack can also be used as a MIA. Namely, given a reconstruction attack $R : \Theta \to \mathcal{Z}$ the adversary outputs $\hat{b} = 1$ if the Neyman-Person criterion accepts the hypothesis $H_0$ and $\hat{b} = 0$ otherwise. We define the false-positive reconstruction rate as the probability of the false-positive outcome in the above game, i.e. $\hat{b} = 1$ and $b = 0$. Similarly, the true-positive reconstruction rate is the probability of the true-positive outcome in the above game, i.e. $\hat{b} = 1$ and $b = 1$.

Algorithm 4 describes the $TPR$ and $FPR$ estimation procedures that we have used to calculate the Neyman-Pearson ROC curves. For explanation of the notation, see Section 2, Section 3 and Algorithm 1 in particular. In the multiclass case we report the class-averaged $TPR$ and $FPR$

$$TPR_{NP} = \frac{1}{C} \sum_{c=0}^{C-1} TPR_{NP}^{(c)}, \quad FPR_{NP} = \frac{1}{C} \sum_{c=0}^{C-1} FPR_{NP}^{(c)}.$$

Figure 8 shows the success rates of our reconstruction attack on a log-scale receiver operating characteristic curve (ROC curve). The ROC curve shows how $TPR$ and $FPR$ changes for all the threshold values $\tau$. In particular, the ROC curve shows how our attack performs at low $FPR$.

**Algorithm 4:** $TPR_{NP}$, $FPR_{NP}$ estimation for reconstructor NN

**Input:** Reconstructor NN $\mathcal{R}$, shadow validation dataset $\mathcal{D}^{val}_{shadow}$, (normalized) error function $\ell$, data prior distribution $\pi_{\mathcal{D}}$, threshold $\tau \in \mathbb{R}$, target reconstruction class $c \in \{0, \ldots, C-1\}$.

**Output:** Reconstruction $TPR^{(c)}_{NP}(\tau)$, $FPR^{(c)}_{NP}(\tau)$.

Initialize $N_{TP} \leftarrow 0$ and $N_{FP} \leftarrow 0$.
Initialize empty lists $L_0 \leftarrow [\,]$ and $L_1 \leftarrow [\,]$.
**foreach** $(\mathcal{D}_k, \theta_k) \in \mathcal{D}^{val}_{shadow}$ **do**
  Sample $\tilde{\mathcal{D}}_k$ from $\pi_{\mathcal{D}}$.
  Select $\mathcal{D}^c_k \subset \mathcal{D}_k$, $\tilde{\mathcal{D}}^c_k \subset \tilde{\mathcal{D}}_k$ as the subsets of images of class $c$.
  Set $\theta^c_k \leftarrow$ the vector $\theta_k$ appended with the one-hot encoded class vector of class $c$.
  $Z_{rec} \leftarrow \mathcal{R}(\theta^c_k)$
  $\ell_{\min} \leftarrow \min_{Z \in \mathcal{D}^c_k} \ell(Z, Z_{rec})$
  $L_0.\text{append}\left(\log \frac{\ell_{\min}}{1 - \ell_{\min}}\right)$
  $\tilde{\ell}_{\min} \leftarrow \min_{Z \in \tilde{\mathcal{D}}^c_k} \ell(Z, Z_{rec})$
  $L_1.\text{append}\left(\log \frac{\tilde{\ell}_{\min}}{1 - \tilde{\ell}_{\min}}\right)$
Estimate the Gaussians $\rho_i = \mathcal{N}(\mu_i, \sigma_i^2)$, $i \in \{0, 1\}$:
$\mu_i \leftarrow \frac{1}{\#L_i} \sum_{\phi \in L_i} \phi$
$\sigma_i^2 \leftarrow \frac{1}{\#L_i} \sum_{\phi \in L_i} (\phi - \mu_i)^2$
$TPR^{(c)}_{NP}(\tau) \leftarrow \# \left\{ \phi \in L_0 : \log\left(\frac{\rho_0(\phi)}{\rho_1(\phi)}\right) > \tau \right\}$
$TPR^{(c)}_{NP}(\tau) \leftarrow \# \left\{ \phi \in L_1 : \log\left(\frac{\rho_0(\phi)}{\rho_1(\phi)}\right) > \tau \right\}$

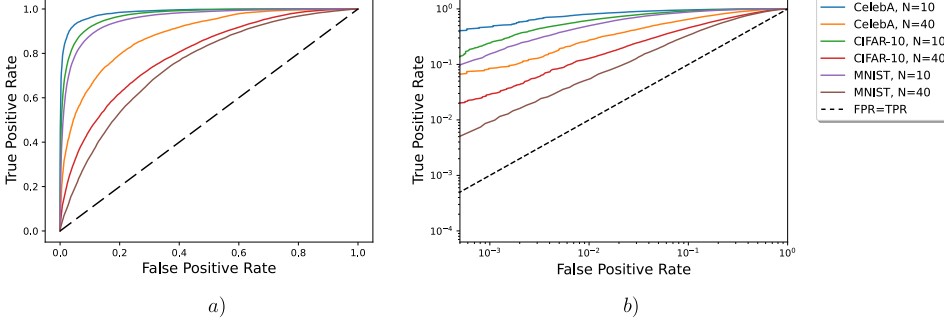

Figure 8: The (Neyman-Pearson) ROC curves for our reconstruction attack for the reconstructor NNs obtained from the experiments described in Section 3. a) Linear scale plot showing the ROC at high $FPR$ and b) log-scale plot showing the ROC at low $FPR$.

### F.1 EVALUATING DATA RECONSTRUCTION ATTACKS VIA HYPOTHESIS TESTING

When deploying a trained reconstructor NN $\mathcal{R} : \Theta \to \mathcal{Z}$, one needs to understand how reliable it is. In Section 2 we have argued that a reconstructor NNs can make up its outputs by generating false-positive reconstructions. In terms of the modified membership security game, given a reconstruction, $\mathcal{R}(\theta)$, and a training set $\mathcal{D}_N$ this raises the problem of deciding whether $\mathcal{R}(\theta)$ is indeed a reconstruction of some element of $\mathcal{D}_N$ or not.

As presented in Section 2, one possible criterion is to consider the random variable

$$\ell_{\min}\left(\mathcal{D}_N, \mathcal{R}(\theta)\right) := \min_{Z \in \mathcal{D}_N} \ell\left(Z, \mathcal{R}(\theta)\right). \tag{9}$$

and the associated true-positive and false-positive reconstruction rates ($TPR$ and $FPR$) via the provably optimal Neyman-Perason criterion (Neyman et al., 1933) (as defined in Section 2) and generate the respective Neyman-Pearson ROC curves which we present in Fig. 8. Alternatively one may consider the more straightforward cumulative criterion (see Section 2) and the resulting cumulative-ROC. In this section, we compare the cumulative ROC curves with the Neyman-Pearson ROC curves. The Neyman-Pearson hypothesis test is also the base for the powerful LiRA membership inference attack (Carlini et al., 2022) which uses similar approach to the one presented here. The Neyman-Pearson ROC curves are provably optimal in the sense that they give the best possible $TPR$ at each fixed $FPR$. To calculate them, we recast the reconstructor NN evaluation problem as hypothesis testing. Namely, we treat the observable $\ell_{\min}\left(\mathcal{D}_N, \mathcal{R}(\theta)\right)$ as a (normalized) random variable supported on $[0; 1]$ which can come from different distributions. More specifically, we study the following hypotheses.

- The null hypothesis $H_0$ states that $\theta$ are the weights of a model trained on $\mathcal{D}_N$ i.e.,

$$H_0: \quad \ell_{\min} = \min_{Z \in \mathcal{D}_N} \ell\left(Z, \mathcal{R}(\theta)\right) \quad \text{with} \quad \mathcal{D}_N \sim \pi_N, \ \theta \sim M(\mathcal{D}_N),$$

  where $M$ is the known randomized training mechanism.

- The alternative hypothesis $H_1$ states that $\theta$ are the weights of a model trained on another training set $\mathcal{D}'_N$ i.e.,

$$H_1: \quad \ell_{\min} = \min_{Z \in \mathcal{D}_N} \ell\left(Z, \mathcal{R}(\theta)\right) \quad \text{with} \quad \mathcal{D}_N \sim \pi_N, \ \mathcal{D}'_N \sim \pi_N, \ \theta \sim M(\mathcal{D}'_N).$$

- The alternative hypothesis $H_1^*$ states that the reconstruction was randomly drawn from the prior distribution $\pi$ i.e.,

$$H_1^*: \quad \ell_{\min} = \min_{Z \in \mathcal{D}_N} \ell\left(Z, Z'\right) \quad \text{with} \quad \mathcal{D}_N \sim \pi_N, \ Z' \sim \pi.$$

Let us denote the respective probability distributions of $\ell_{\min}$ as $\rho_0$, $\rho_1$ and $\rho_1*$. When deciding between $H_0$ and $H_1$, the Neyman-Pearson hypothesis testing criterion accepts $H_0$ for a given value of $\ell_{\min}$ when the likelihood-ratio

$$\log\left(\frac{\rho_0(\ell_{\min})}{\rho_1(\ell_{\min})}\right) > C \tag{10}$$

for some threshold value $C \in \mathbb{R}$. The $TPR$ in this test is the probability of accepting $H_0$ for $\ell_{\min} \sim \rho_0$ and the $FPR$ the probability of accepting $H_0$ for $\ell_{\min} \sim \rho_1$. Analogous criteria are applied when deciding between $H_0$ and $H_1^*$. The Neyman-Pearson ROC curves are obtained by varying the threshold $C$. Unfortunately, in our case we do not have access to the analytic expressions for $\rho_0$, $\rho_1$, $\rho_1*$, so we cannot calculate the likelihoods in Equation 10 exactly. In practice, we will approximate the distributions $\rho_0$, $\rho_1$ and $\rho_1*$ by something more tractable. To this end, we will look at the distribution of the random variable

$$\phi = \log\left(\frac{\ell_{\min}}{1 - \ell_{\min}}\right) \in \mathbb{R}.$$

This is because our experimental evidence shows that $\phi$ is approximately normally distributed, as opposed to $\ell_{\min}$. The relevant histograms for the CelebA-reconstructor NN are shown in Fig. 9. The histograms have two important properties: i) the separation between the distribution of $\ell_{\min}$ corresponding to $H_0$ and $H_1$ or $H_1^*$ decreases with $N$ making it more difficult to distinguish between the hypotheses, ii) there is more separation between the distributions of $\ell_{\min}$ corresponding to $H_0$ and

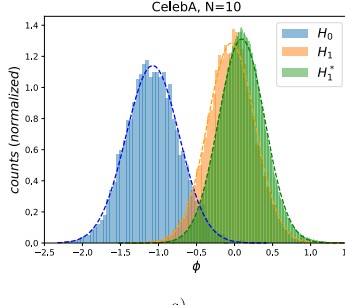 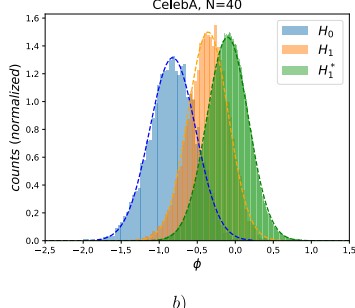

$a)$ $b)$

Figure 9: The histograms illustrating the approximately normal distributions of $\phi$ for the different hypotheses. The sample sizes were $10^4$, $10^4$ and $10^6$ for $H_0$, $H_1$ and $H_1^*$ respectively. The dashed curves show the Gaussians fitted according the maximum likelihood.

Table 5: The parameters of the Gaussians $\rho \propto e^{-(\phi-\mu)^2/2\sigma^2}$ fitted in Fig. 9. Note that the fitted values of $\sigma$ for the different hypotheses are close to each other. By Theorem 2 this explains the close similarity between the Neyman-Pearson ROCs and the cumulative ROCs presented in Fig. 10 and Fig. 11.

|  | $N = 10$ | | $N = 40$ | |
| --- | --- | --- | --- | --- |
| Hypothesis | $\mu$ | $\sigma$ | $\mu$ | $\sigma$ |
| $H_0$ | $-1.076$ | $0.349$ | $-0.823$ | $0.303$ |
| $H_1$ | $-0.043$ | $0.310$ | $-0.357$ | $0.266$ |
| $H_1^*$ | $0.093$ | $0.304$ | $-0.101$ | $0.273$ |

$H_1^*$ than between $H_0$ and $H_1$ showing that the reconstructor $NN$ is better at generating false-positive reconstructions than simple random guessing.

In Fig 10 we compare the cumulative- and the Neyman-Pearson ROC curves for the reconstructor NNs trained in the CelebA-experiment. The plots show that both ROC curves are extremely close to each other. Below, we prove that under certain conditions they are exactly the same and derive an analytic formula for these ROC curves. The analytic formula is compared with our experimental ROC curves in Fig. 11.

**Theorem 2.** Let $\rho_0$ and $\rho_1$ be the distributions of the observable $\omega$ supported on a connected domain $\Omega \subset \mathbb{R}$ under hypotheses $H_0$ and $H_1$ respectively. Define the cumulative- and the Neyman-Pearson ROC via

$$TPR_{cum}(\tau) := \mathcal{P}_{\omega \sim \rho_0}\left[\omega < \tau\right], \quad FPR_{cum}(\tau) := \mathcal{P}_{\omega \sim \rho_1}\left[\omega < \tau\right],$$
$$TPR_{NP}(C) := \mathcal{P}_{\omega \sim \rho_0}\left[\rho_0(\omega) > C\rho_1(\omega)\right], \quad FPR_{NP}(C) := \mathcal{P}_{\omega \sim \rho_1}\left[\rho_0(\omega) > C\rho_1(\omega)\right].$$

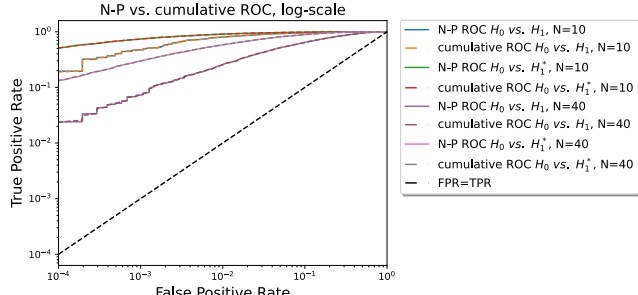

Figure 10: The comparison between the cumulative- and Neyman-Pearson (N-P) ROC curves for different pairs of hypotheses in the CelebA-experiment. There cumulative- and N-P ROC overlap almost exactly. See Theorem 2 for an explanation of this effect.

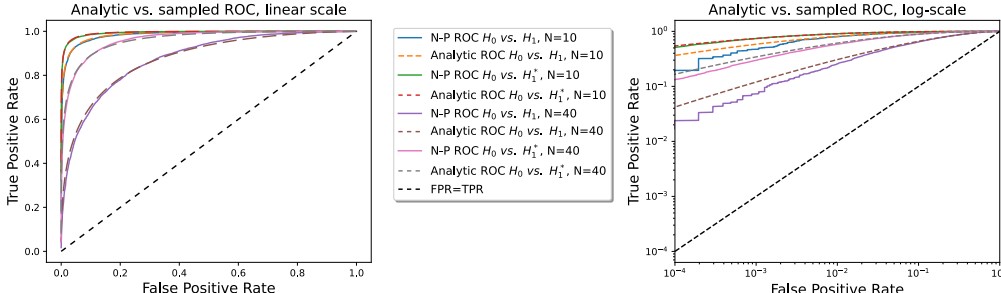

Figure 11: The comparison between the analytic ROC described by Equation 11 and Neyman-Pearson (N-P) ROC curves for different pairs of hypotheses in the CelebA-experiment. There is a good agreement between the analytic expressions and the experiment, but the curves often diverge a bit at low $FPR$. We suppose this may be due to insufficient sampling at this extreme of the ROCs.

Assume that there exists a strictly increasing differentiable transformation $\Phi : \Omega \to \mathbb{R}$ such that

$$\Phi(\omega) \sim \mathcal{N}\left(\mu_0, \sigma_0^2\right) \quad \text{if} \quad \omega \sim \rho_0, \quad \Phi(\omega) \sim \mathcal{N}\left(\mu_1, \sigma_1^2\right) \quad \text{if} \quad \omega \sim \rho_1$$

with $\mu_0 < \mu_1$.

The following facts hold.

1. If $\sigma_0 = \sigma_1$, the cumulative ROC curve and the Neyman-Pearson ROC curve are identical and given by the following formula

$$TPR = \frac{1}{2} - \frac{1}{2}\operatorname{erf}\left(\frac{\mu_0 - \mu_1}{\sqrt{2}\sigma_0} + \frac{\sigma_1}{\sigma_0}\operatorname{erf}^{-1}\left(1 - 2FPR\right)\right), \tag{11}$$

   where $\operatorname{erf}$ is the error function.

2. The Neyman-Pearson ROC curve is a continuous function of $\sigma_0$ and $\sigma_1$, thus when $\sigma_0$ is close to $\sigma_1$, the Neyman-Pearson ROC curve remains close to the cumulative ROC curve.

3. For arbitrary $\sigma_1, \sigma_0$ the Neyman-Pearson ROC is asymptotically equivalent to the cumulative ROC described by Equation 11 in the low-$FPR$ regime.

*Proof.* Since the transformation $\Phi$ is strictly increasing, we get that $\omega < \tau$ if and only if $\Phi(\omega) < \Phi(\tau)$. Thus,

$$TPR_{cum}(\tau) := \mathcal{P}_{\phi \sim \mathcal{N}\left(\mu_0, \sigma_0^2\right)}\left[\phi < \Phi(\tau)\right], \quad FPR_{cum}(\tau) := \mathcal{P}_{\phi \sim \mathcal{N}\left(\mu_1, \sigma_1^2\right)}\left[\phi < \Phi(\tau)\right]. \tag{12}$$

The above probabilities can be calculated analytically.

$$TPR_{cum}(\tau) := \frac{1}{2}\left(1 - \operatorname{erf}\left(\frac{\mu_0 - \Phi(\tau)}{\sqrt{2}\sigma_0}\right)\right), \tag{13}$$

$$FPR_{cum}(\tau) := \frac{1}{2}\left(1 - \operatorname{erf}\left(\frac{\mu_1 - \Phi(\tau)}{\sqrt{2}\sigma_1}\right)\right). \tag{14}$$

It is a matter of a straightforward calculation to extract $\Phi(\tau)$ from Equation 14 and plug it into Equation 13 to obtain the cumulative ROC curve given by Equation 11. In the remaining part of this proof we compare the ROC curve given by Equation 11 with the ROC curve is obtained from the Neyman-Pearson criterion.

The transformation $\Phi$ is differentiable and strictly increasing, so $\left(\Phi^{-1}\right)'(\phi) > 0$ for all $\phi \in \mathbb{R}$. Thus, $\rho_0(\omega) > C\rho_1(\omega)$ for some $\omega = \Phi^{-1}(\phi)$ if and only if

$$\rho_0\left(\Phi^{-1}(\phi)\right)\left(\Phi^{-1}\right)'(\phi) > C\rho_1\left(\Phi^{-1}(\phi)\right)\left(\Phi^{-1}\right)'(\phi).$$

Recall that the transformation $\Phi$ is chosen so that

$$\rho_i\left(\Phi^{-1}(\phi)\right)\left(\Phi^{-1}\right)'(\phi) = \tilde{\rho}_i(\phi) := \frac{1}{\sqrt{2\pi}\sigma_i}\exp\left(-\frac{(\phi - \mu_i)^2}{2\sigma_i^2}\right), \quad i = 0, 1.$$

Thus, using the monotonicity of the logarithm we can write equivalently that

$$TPR_{NP}(\tilde{C}) := \mathcal{P}_{\phi \sim \tilde{\rho}_0}\left[\log \frac{\tilde{\rho}_0(\phi)}{\tilde{\rho}_1(\phi)} > \tilde{C}\right], \quad FPR_{NP}(\tilde{C}) := \mathcal{P}_{\phi \sim \tilde{\rho}_1}\left[\log \frac{\tilde{\rho}_0(\phi)}{\tilde{\rho}_1(\phi)} > \tilde{C}\right]. \quad (15)$$

Plugging in the Gaussian form of $\tilde{\rho}_0$ and $\tilde{\rho}_1$ we get that the log-likelihood-ratio is given by the following equation.

$$\log \frac{\tilde{\rho}_0(\phi)}{\tilde{\rho}_1(\phi)} = \log \frac{\sigma_1}{\sigma_0} - \frac{(\phi - \mu_0)^2}{2\sigma_0^2} + \frac{(\phi - \mu_1)^2}{2\sigma_1^2}. \quad (16)$$

In what follows, we will assume that $\sigma_0 \geq \sigma_1$ which is the case in the CelebA experiments. The case $\sigma_0 < \sigma_1$ can be treated fully analogously. Let us start with the case when $\sigma_0 = \sigma_1 \equiv \sigma$. Then,

$$\log \frac{\tilde{\rho}_0(\phi)}{\tilde{\rho}_1(\phi)} = -\frac{\mu_1 - \mu_0}{\sigma^2}\phi + \frac{(\mu_1 + \mu_0)(\mu_1 - \mu_0)}{2\sigma^2},$$

i.e., the log-likelihood-ratio is a negatively-sloped linear function of $\phi$. Thus, $\log \frac{\tilde{\rho}_0(\phi)}{\tilde{\rho}_1(\phi)} > \tilde{C}$ whenever $\phi < \phi_0(\tilde{C})$ where

$$\phi_0(\tilde{C}) = \frac{(\mu_1 + \mu_0)(\mu_1 - \mu_0) - 2\sigma^2\tilde{C}}{2(\mu_1 - \mu_0)}. \quad (17)$$

According to Equation 15 this yields

$$\overline{TPR}_{NP}\left(\tilde{C}\right) = \frac{1}{\sqrt{2\pi}\sigma}\int_{-\infty}^{\phi_0} \exp\left(-\frac{(\phi - \mu_0)^2}{2\sigma^2}\right) d\phi = \frac{1}{2}\left(1 - \mathrm{erf}\left(\frac{\mu_0 - \phi_0(\tilde{C})}{\sqrt{2}\sigma}\right)\right),$$

$$\overline{FPR}_{NP}\left(\tilde{C}\right) = \frac{1}{\sqrt{2\pi}\sigma}\int_{-\infty}^{\phi_0} \exp\left(-\frac{(\phi - \mu_1)^2}{2\sigma^2}\right) d\phi = \frac{1}{2}\left(1 - \mathrm{erf}\left(\frac{\mu_1 - \phi_0(\tilde{C})}{\sqrt{2}\sigma}\right)\right). \quad (18)$$

The Equations Equation 18 become identical with the Equations Equation 13, Equation 14 describing the cumulative $TPR$ and $FPR$ under the identification $\phi_0(\tilde{C}) = \Phi(\tau)$. Such an identification is indeed possible – using Equation 17 for every $\nu \in \Omega$ one can find a unique $\tilde{C}(\nu)$ for which $\phi_0(\tilde{C}) = \Phi(\tau)$. Thus, we have shown that the Neyman-Pearson ROC is equal to the cumulative ROC if $\sigma_0 = \sigma_1$.

Let us next prove the statements 2. and 3. of this Theorem concerning the case when $\sigma_0 > \sigma_1$. Then, the log-likelihood-ratio Equation 16 is described by a convex parabola. The Neyman-Pearson criterion leads to two roots whenever

$$\tilde{C} > \tilde{C}_{\min} = \log \frac{\sigma_1}{\sigma_0} - \frac{1}{2}\frac{(\mu_0 - \mu_1)^2}{\sigma_0^2 - \sigma_1^2}.$$

The two roots are of the form $r_0 \pm \delta$, where

$$r_0 = \frac{\mu_1 \sigma_0^2 - \mu_0 \sigma_1^2}{\sigma_0^2 - \sigma_1^2}, \quad \delta\left(\tilde{C}\right) = \frac{\sigma_0 \sigma_1 \sqrt{(\mu_0 - \mu_1)^2 + 2\left(\tilde{C} + \log \frac{\sigma_0}{\sigma_1}\right)(\sigma_0^2 - \sigma_1^2)}}{\sigma_0^2 - \sigma_1^2} > 0. \quad (19)$$

Hence, we can compute

$$TPR_{NP}\left(\tilde{C}\right) = 1 - \mathcal{P}_{\phi \sim \tilde{\rho}_0}\left[\log \frac{\tilde{\rho}_0(\phi)}{\tilde{\rho}_1(\phi)} < \tilde{C}\right] = 1 - \frac{1}{\sqrt{2\pi}\sigma_0}\int_{r_0 - \delta}^{r_0 + \delta} \exp\left(-\frac{(\phi - \mu_0)^2}{2\sigma_0^2}\right) d\phi$$

$$= 1 - \frac{1}{2}\left(\mathrm{erf}\left(\frac{\mu_0 - r_0 + \delta\left(\tilde{C}\right)}{\sqrt{2}\sigma_0}\right) - \mathrm{erf}\left(\frac{\mu_0 - r_0 - \delta\left(\tilde{C}\right)}{\sqrt{2}\sigma_0}\right)\right)$$

$$(20)$$

Similarly, we get

$$FPR_{NP}\left(\tilde{C}\right) = 1 - \frac{1}{2}\left(\mathrm{erf}\left(\frac{\mu_1 - r_0 + \delta\left(\tilde{C}\right)}{\sqrt{2}\sigma_1}\right) - \mathrm{erf}\left(\frac{\mu_1 - r_0 - \delta\left(\tilde{C}\right)}{\sqrt{2}\sigma_1}\right)\right). \quad (21)$$

Clearly, the above $TPR_{NP}$ and $FPR_{NP}$ are continuous functions of $\sigma_0, \sigma_1$ when $\sigma_0 > \sigma_1$. Let us next show that they have a finite limit when $\sigma_1 \to \sigma_0$ from below and that this limit again describes the cumulative ROC curve. This result means that the Neyman-Pearson ROC curve is a continuous function of $\sigma_0, \sigma_1$ also on the line $\sigma_1 = \sigma_0$. Thus, when $\sigma_1 = \sigma_0 - \epsilon$ with $\epsilon$ small the resulting Neyman-Pearson ROC curve will remain close to the cumulative ROC curve by continuity arguments. Using the expressions Equation 19 we find the following limits.

$$\mu_0 - r_0 + \delta\left(\tilde{C}\right) \xrightarrow{\sigma_1 \to \sigma_0^-} \frac{1}{2}\left(\mu_1 - \mu_0 + \frac{2\sigma^2 \tilde{C}}{\mu_1 - \mu_0}\right), \quad \mu_0 - r_0 - \delta\left(\tilde{C}\right) \xrightarrow{\sigma_1 \to \sigma_0^-} -\infty,$$

$$\mu_1 - r_0 + \delta\left(\tilde{C}\right) \xrightarrow{\sigma_1 \to \sigma_0^-} \frac{1}{2}\left(\mu_0 - \mu_1 + \frac{2\sigma^2 \tilde{C}}{\mu_1 - \mu_0}\right), \quad \mu_1 - r_0 - \delta\left(\tilde{C}\right) \xrightarrow{\sigma_1 \to \sigma_0^-} -\infty.$$

After using the fact that $\mathrm{erf}(u) \xrightarrow{u \to -\infty} -1$ and some algebra we get the following limits for $TPR_{NP}$ and $FPR_{NP}$.

$$TPR_{NP}\left(\tilde{C}\right) \xrightarrow{\sigma_1 \to \sigma_0^-} \overline{TPR}_{NP}\left(\tilde{C}\right), \quad FPR_{NP}\left(\tilde{C}\right) \xrightarrow{\sigma_1 \to \sigma_0^-} \overline{FPR}_{NP}\left(\tilde{C}\right),$$

where $\overline{TPR}_{NP}$ and $\overline{FPR}_{NP}$ were given in Equation 18 and describe the cumulative ROC curve.

It remains to prove statement 3. concerning the asymptotic equivalence of the Neyman-Pearson ROC and the cumulative ROC curves at low $FPR$. Note that the low-$FPR$ limit is equivalent to taking $\tilde{C} \to \infty$ in Equations Equation 20 and Equation 21. Using the analytic formula for the cumulative ROC curve Equation 11 we get that the asymptotic equivalence of the Neyman-Pearson ROC and the cumulative ROC curves means that

$$\frac{1}{2}\frac{1 - \mathrm{erf}\left(\frac{\mu_0 - \mu_1}{\sqrt{2}\sigma_0} + \frac{\sigma_1}{\sigma_0}\mathrm{erf}^{-1}\left(1 - 2FPR_{NP}\left(C\right)\right)\right)}{TPR_{NP}\left(C\right)} \xrightarrow{C \to \infty} 1. \tag{22}$$

To prove Equation 22 we will make use of the asymptotic expansions of the error function and the inverse error function. Using the asymptotic expansion (Olver, 1997)

$$\mathrm{erf}(u) \sim \mathrm{sgn}(u)\left(1 - e^{-u^2}\left(\frac{1}{u\sqrt{\pi}} + \mathcal{O}\left(u^{-2}\right)\right)\right), \quad |u| \to \infty$$

we get the following asymptotic expansions for $TPR_{NP}\left(C\right)$ and $FPR_{NP}\left(C\right)$ when $C \to \infty$

$$TPR_{NP}\left(C\right) \sim e^{-\left(\frac{\mu_0 - r_0 + \delta(C)}{\sqrt{2}\sigma_0}\right)^2}\left(\frac{\sigma_0}{\sqrt{2\pi}}\frac{1}{\delta(C)} + \mathcal{O}\left(\delta(C)^{-2}\right)\right),$$

$$FPR_{NP}\left(C\right) \sim e^{-\left(\frac{\mu_1 - r_0 + \delta(C)}{\sqrt{2}\sigma_1}\right)^2}\left(\frac{\sigma_1}{\sqrt{2\pi}}\frac{1}{\delta(C)} + \mathcal{O}\left(\delta(C)^{-2}\right)\right). \tag{23}$$

Our next goal is to find the asymptotic expansion of the numerator of the LSH of Equation 22. To this end, we use the following asymptotic expansion of the inverse error function

$$\mathrm{erf}^{-1}\left(1 - \frac{e^{-u^2}}{\sqrt{\pi}u}\right) \sim u + \mathcal{O}\left(u^{-2}\right), \quad u \to \infty. \tag{24}$$

The formula Equation 24 follows from the standard expansion of $\mathrm{erf}^{-1}(x)$ around $x \to 1$ (see Equation (2) in Blair et al. (1976))

$$\mathrm{erf}^{-1}\left(x\right) \sim \sqrt{\eta} - \frac{1}{4}\log(\eta)\eta^{-1/2} + \mathcal{O}\left(\eta^{-3/2}\right), \quad \eta = -\log(\sqrt{\pi}(1 - x)), \quad x \to 1. \tag{25}$$

The formula Equation 24 is obtained from Equation 25 by putting $\eta = u^2 + \log u$ and making use of the facts that

$$\sqrt{u^2 + \log u} \sim u + \frac{1}{2}\frac{\log u}{u} + \mathcal{O}\left(u^{-2}\right), \quad \frac{1}{4}\frac{\log(u^2 + \log u)}{\sqrt{u^2 + \log u}} \sim \frac{1}{2}\frac{\log u}{u} + \mathcal{O}\left(u^{-2}\right), \quad u \to \infty.$$

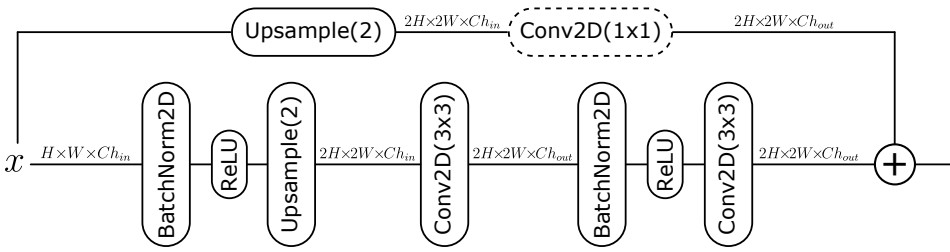

Figure 12: The residual block of the reconstructor NN. The "+"-operation means element-wise addition.

Thus, by putting $u = \frac{-\mu_1 + r_0 + \delta(C)}{\sqrt{2}\sigma_1}$ in Equation 24 we have that at $C \to \infty$

$$\operatorname{erf}^{-1}\left(1 - 2FPR_{NP}(C)\right) \sim \frac{\mu_1 - r_0 + \delta(C)}{\sqrt{2}\sigma_1} + \mathcal{O}\left(\delta(C)^{-2}\right)$$

After plugging the above expansion into the numerator of the LHS of Equation 22 followed by some straightforward algebra we obtain that at $C \to \infty$

$$\frac{1}{2} - \frac{1}{2}\operatorname{erf}\left(\frac{\mu_0 - \mu_1}{\sqrt{2}\sigma_0} + \frac{\sigma_1}{\sigma_0}\operatorname{erf}^{-1}(1 - 2FPR_{NP}(C))\right) \sim$$

$$\sim \frac{1}{2} - \frac{1}{2}\operatorname{erf}\left(\frac{\mu_0 - r_0 + \delta(C)}{\sqrt{2}\sigma_0} + \mathcal{O}\left(\delta(C)^{-2}\right)\right)$$

$$\sim e^{-\left(\frac{\mu_0 - r_0 + \delta(C)}{\sqrt{2}\sigma_0}\right)^2}\left(\frac{\sigma_0}{\sqrt{2\pi}}\frac{1}{\delta(C)} + \mathcal{O}\left(\delta(C)^{-2}\right)\right)$$

which in the leading order is identical to the asymptotic expansion of $TPR_{NP}(C)$ from Equation 23. This proves Equation 22. $\qquad\square$

## G   RECONSTRUCTOR NN ARCHITECTURE AND TRAINING

The reconstructor is a residual network. Each residual block uses ReLU activations and consists of: 1) 2D batch norm layer followed by ReLU whose output is upsampled via the nearest-neighbours algorithm, 2) a pre-activation $(3 \times 3)$-convolutional layer with 1-padding and stride 1 followed by another 2D batch norm layer and ReLU, 3) another pre-activation $(3 \times 3)$-convolutional layer with 1-padding and stride 1. The bypass connection contains upsampling via the nearest-neighbours algorithm and, if the number of output channels of the residual block is different than the number of its input channels, then the bypass-upsampling is followed by a pre-activation $(1 \times 1)$-convolution layer with 0-padding and stride 1. See Fig. 12. This residual block has the same architecture as the one used in (Gulrajani et al., 2017; Wu et al., 2020), except for the bypass-convolution which we need to allow for the number of channels to change. Table 6 shows the full architecture which is also slightly modified relative to the original version from (Gulrajani et al., 2017; Wu et al., 2020) – the first layer is convolutional and we change the number of channels between some of the consecutive residual blocks. For MNIST and CIFAR the output image has $32 \times 32$ resolution while for CelebA the output image has $64 \times 64$ resolution. For further details, please refer to our open-source implementation on GitHub:

- Training Data Reconstruction Attacks `https://anonymous.4open.science/r/ training-data-reconstruction-1EB2`.

In Figure 13 we summarize the reconstructor-NN training process.

## H   FURTHER RECONSTRUCTION EXAMPLES

In this section, we present the following reconstruction examples. In all the examples successful reconstructions are those for which the Neyman-Pearson criterion yields $FPR = 1\%$. The successful

Table 6: Summary of the reconstructor NN architecture. $D_\Theta$ is the number of trainable parameters in the attacked model that are accessed by the reconstructor. We have $D_\Theta$ equal to 2570, 12810 and 8201 for the MNIST-, CIFAR- and CelebA-experiments respectively. $S$ is reconstructor's internal size parameter. In the MNIST-experiment we take $S = 256$ and in the CIFAR- and CelebA-experiments we take $S = 512$. $Ch_{out} = 1$ for MNIST and $Ch_{out} = 3$ for CIFAR, CelebA.

| Reconstructor $R(\theta)$ | | | |
|---|---|---|---|
| | Kernel Size | Output Shape | Notes |
| $\theta$ | – | $1 \times 1 \times (D_\Theta + 10)$ | – |
| ConvTranspose2D | $4 \times 4$ | $4 \times 4 \times 4S$ | $str. = 1, pad. = 0$ |
| Residual Block | $3 \times 3$ | $8 \times 8 \times 2S$ | – |
| Residual Block | $3 \times 3$ | $16 \times 16 \times S$ | – |
| Residual Block | $3 \times 3$ | $32 \times 32 \times S$ | – |
| Residual Block | $3 \times 3$ | $64 \times 64 \times S$ | only for CelebA |
| BatchNorm2D | – | $32 \times 32 \times S/64 \times 64 \times S$ | – |
| ReLU | – | $32 \times 32 \times S/64 \times 64 \times S$ | – |
| Conv2D | $3 \times 3$ | $32 \times 32 \times Ch_{out}/64 \times 64 \times Ch_{out}$ | $str. = 1, pad. = 1$ |
| Tanh | – | $32 \times 32 \times Ch_{out}/64 \times 64 \times Ch_{out}$ | – |

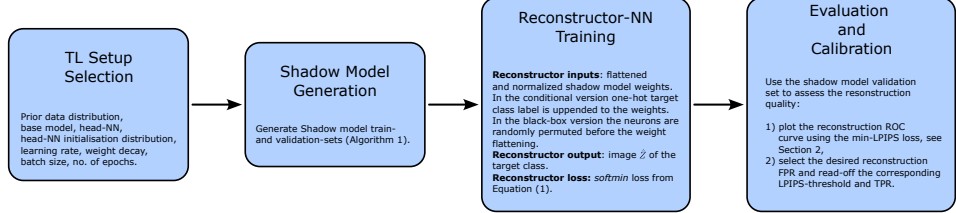

Figure 13: Flowchart summarizing the reconstructor-NN training process.

reconstructions are marked by the green tick-signs. Top rows show original images and bottom rows the reconstructions.

- Figures 14 and 15 show white-box reconstruction examples for the CelebA experiment with $N = 10$ and $N = 40$ respectively. Each training set contains $N/10$ images of positive class corresponding to a given person. The bottom rows contain the reconstructions and the top rows contain their closest match out of the $N/10$ images of the positive class from the original training set.

- Figure 16 shows white-box reconstruction examples from the conditional reconstructor for the MNIST experiment with $N = 10$ and $N = 40$. Every training set contains $N/10$ images of each class. The bottom rows contain the reconstructions and the top rows contain their closest match from the original training set.

- Figure 17 shows white-box reconstruction examples from the conditional reconstructor for the CIFAR experiment with $N = 10$ and $N = 40$. Every training set contains $N/10$ images of each class. The bottom rows contain the reconstructions and the top rows contain their closest match from the original training set. Note that the reconstructions in Fig. 17b) are more generic and often lack details. This is due to the low reconstruction $TPR$ for $N = 40$ (see Table 2).

- Figure 18 shows examples of false-positive white-box reconstructions for MNIST and CIFAR-10. The bottom rows contain the random outputs of the conditional reconstructor NN and the top rows contain their closest match from the original (random) training set of the size $N = 40$.

- Figure 19 shows white-box reconstruction examples in four-shot learning illustrating that the reconstructor-NN outputs specific examples from the original training set rather than their aggregates.

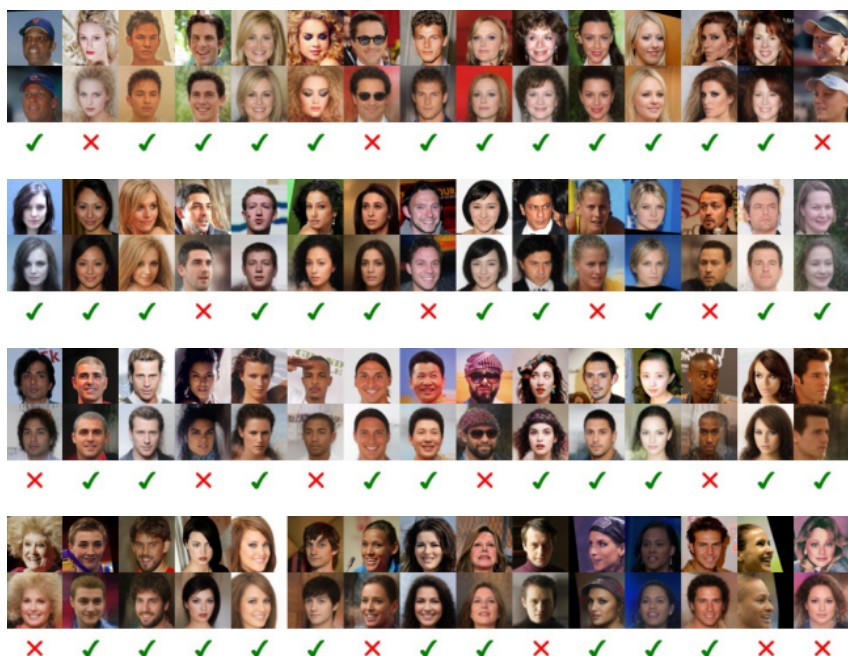

Figure 14: White-box reconstruction examples for the CelebA experiment with $N = 10$. Top rows show original images and bottom rows the reconstructions.

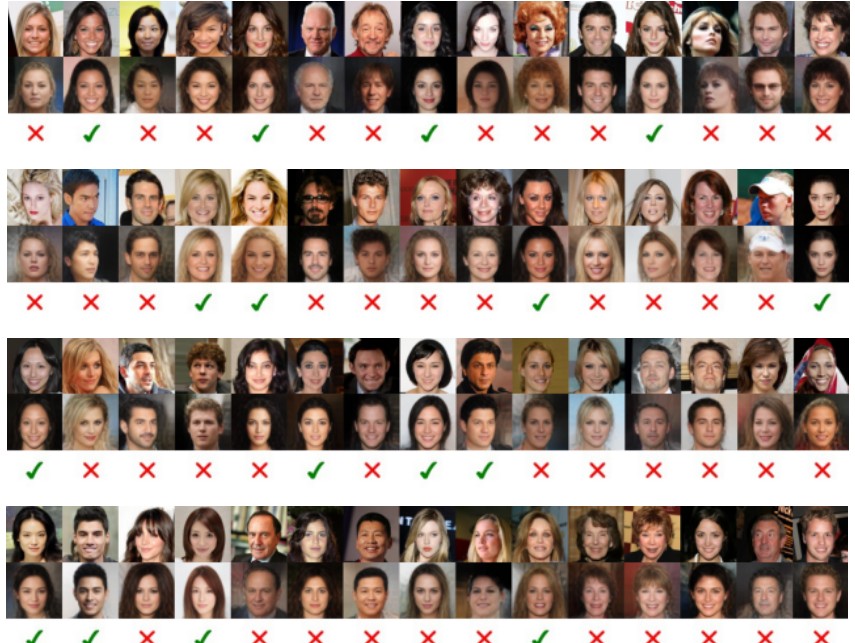

Figure 15: White-box reconstruction examples for the CelebA experiment with $N = 40$. Top rows show original images and bottom rows the reconstructions.

- Figure 20 shows black-box reconstruction examples from one-shot TL in the MNIST, CIFAR-10 (conditional reconstructor) and CelebA (unconditional reconstructor) setups.

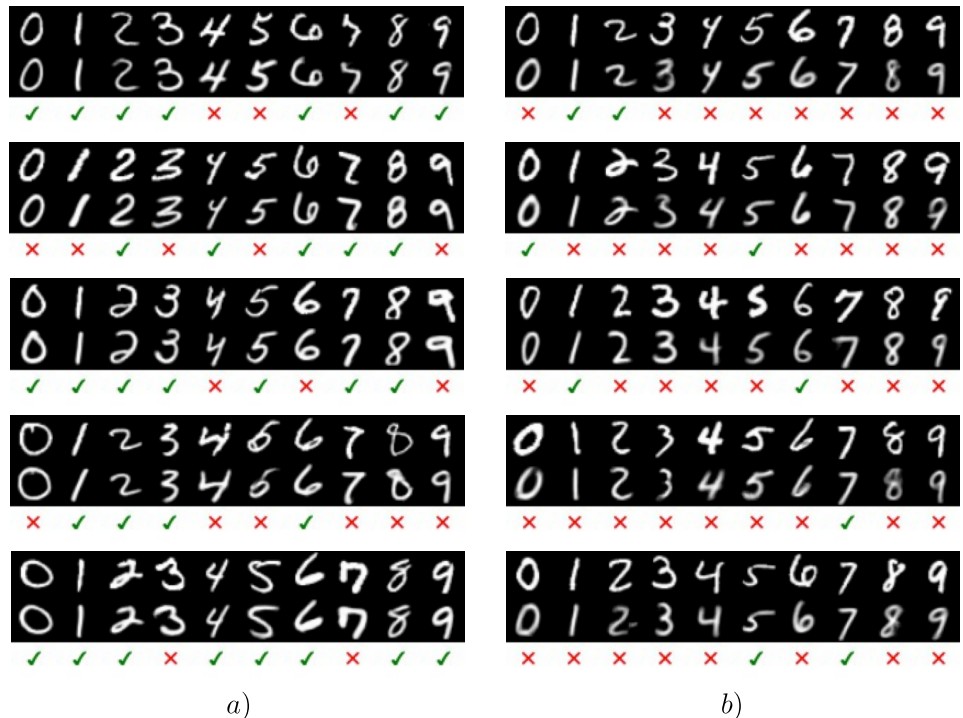

Figure 16: White-box reconstruction examples from the conditional reconstructor for the MNIST experiment with a) $N = 10$ and b) $N = 40$. Top rows show original images and bottom rows the reconstructions.

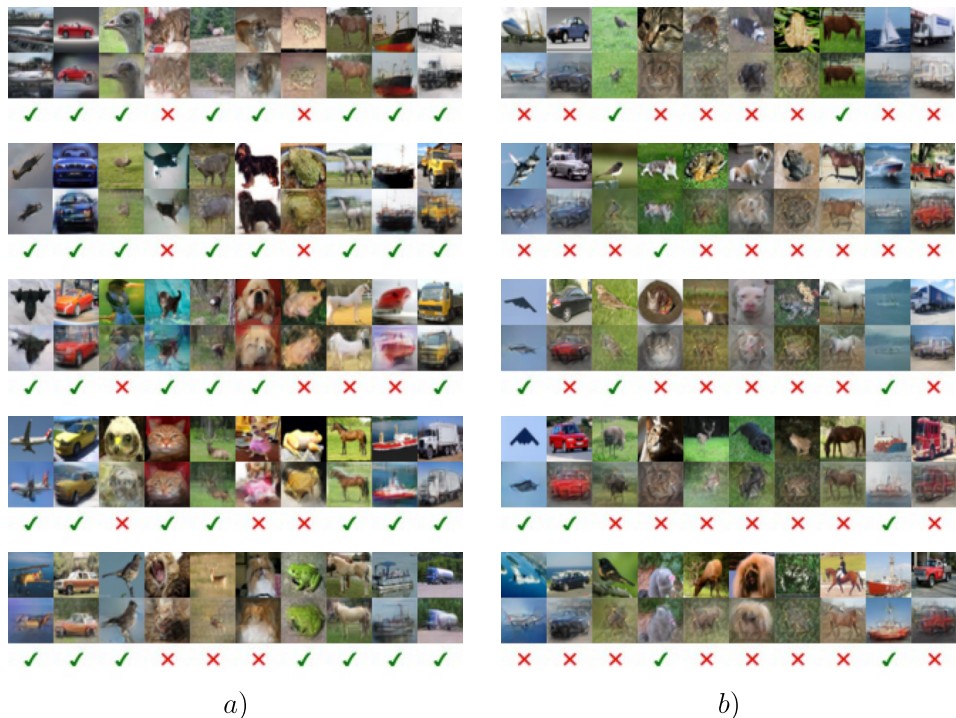

Figure 17: White-box reconstruction examples from the conditional reconstructor for the CIFAR experiment with a) $N = 10$ and b) $N = 40$. Top rows show original images and bottom rows the reconstructions.

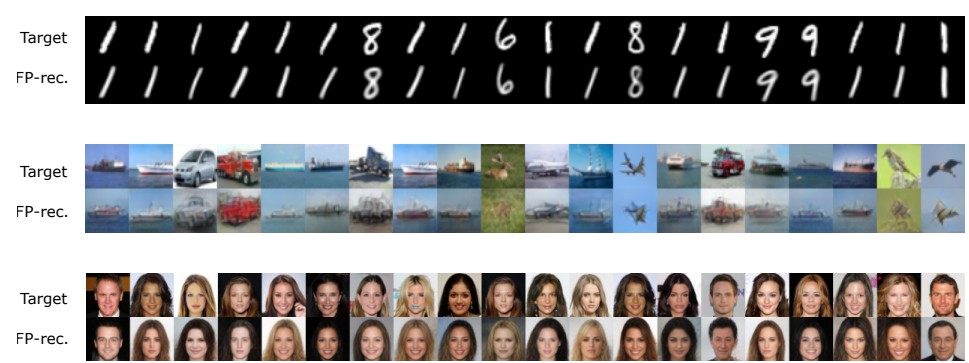

Figure 18: Examples of false-positive (white-box) reconstructions for MNIST, CIFAR-10 and CelebA ($N = 40$) at the LPIPS threshold corresponding to $FPR = 1\%$. Top rows show original images and bottom rows the reconstructions.

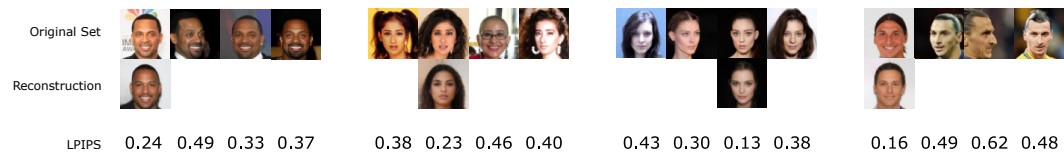

Figure 19: Target set and white-box reconstruction pairs in four-shot learning in the CelebA-experiment. The numbers show the corresponding pairwise LPIPS-error values. The LPIPS threshold corresponding to the (cumulative) $FPR = 1\%$ is equal to 0.266. The reconstructor-NN outputs specific examples from the original training set rather than their aggregates.

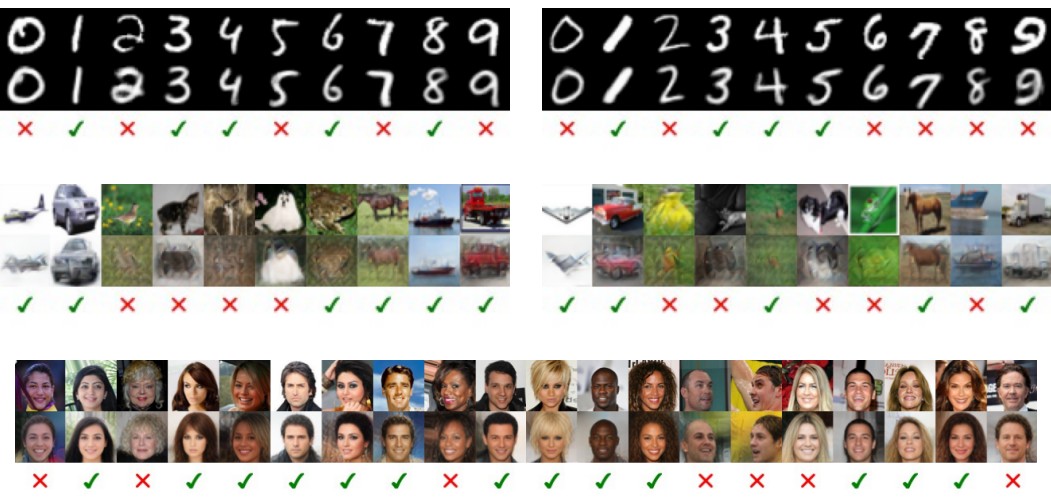

Figure 20: Hard-label black-box reconstruction examples from one-shot (i.e. $N = 10$) TL in the MNIST, CIFAR-10 and CelebA setups. Top rows show original images and bottom rows the reconstructions.

## I  TECHNICAL DETAILS OF THE EXPERIMENTAL SETUPS

In this section we present details of the experiments presented in Section 3. For implementation details please refer to our code on GitHub:

- Training Data Reconstruction Attacks `https://anonymous.4open.science/r/training-data-reconstruction-1EB2`.

### I.1  IMAGE CLASSIFIER PRE-TRAINING

Below, we specify the pre-training steps that we have done in each of the experiments. The goal is to pre-train the base models on a sufficiently general task so that they develop robust features which can be successfully transferred to other more specialized classification tasks where much smaller amounts of data are available.

1. **MNIST**  We pre-train a scaled-down VGG-11 neural net (Simonyan & Zisserman, 2015) on the EMNIST-Letters (Cohen et al., 2017) classification task: $128K$ data, 26 classes. Our implemented VGG-11 differs from the original one in (Simonyan & Zisserman, 2015) only by the number of channels which we divide by the factor of 16 (e.g. the first $(3 \times 3)$-convolution has 4 output channels instead of the original 64). Similarly, the sizes of the final three fully connected layers are $256 - 256 - 26$ instead of the original $4096 - 4096 - 1000$ that has been designed for the ImageNet-1K classification. We initialize the weights of our VGG-11 randomly, rescale the input images to the $32 \times 32$ resolution and normalize them according to $X \rightarrow X/127.5 - 1$. We train for 50 epochs with the learning rate of $10^{-4}$ and mini-batch size 256 using Adam optimizer and cross-entropy loss. We obtain the base model test accuracy of $94.6\%$.

2. **CIFAR**  We pre-train EfficientNet-B0 (Tan & Le, 2019) on the CIFAR-100 (Krizhevsky, 2009) classification task: $50K$ data, 100 classes. We use the implementation of EfficientNet-B0 provided by Torchvision (Marcel & Rodriguez, 2010). For the weight initialization we use the weights that were pre-trained on the ImageNet-1K classification (available in Torchvision). We subsequently remove the 1000 output neurons from the top fully connected layer and replace them with 100 neurons that we initialize with Pytorch's default initialization. The input images are resized to the $224 \times 224$ resolution using the bicubic interpolation. The pixel values are subsequently rescaled to $[0, 1]$ and normalized as $X \rightarrow (X - \mu)/\sigma$, where $\mu = [0.485, 0.456, 0.406]$ and $\sigma = [0.229, 0.224, 0.225]$ for each channel. We pre-train in two stages. In both stages we use the Adam optimizer and the cross-entropy loss. In the first stage we freeze all the parameters except for the parameters of the output layer and train the output layer for 20 epochs with learning rate $10^{-4}$, weight decay $10^{-4}$ and batch size 256. In the second stage we unfreeze all the parameters except for the batch-norm layers and train the entire network for 200 epochs with learning rate $10^{-6}$ and batch size 64. We obtain the base model test accuracy of $85.9\%$ which is close to some of the reported benchmarks for EfficientNet-B0, see Tan & Le (2019).

3. **CelebA**  We use WideResNet-50 (Zagoruyko & Komodakis, 2016) pre-trained on ImageNet-1K. We use the implementation of WideResNet-50-2 and the pre-trained weights provided by Torchvision (Marcel & Rodriguez, 2010). The input images are resized to the $232 \times 232$ resolution using the bicubic interpolation and center-cropped to the size $224 \times 224$. The pixel values are subsequently rescaled to $[0, 1]$ and normalized as $X \rightarrow (X - \mu)/\sigma$, where $\mu = [0.485, 0.456, 0.406]$ and $\sigma = [0.229, 0.224, 0.225]$ for each channel.

Note that we could have skipped the above pre-training with CIFAR-100 and just use the weights pre-trained on ImageNet-1K. However, note that our goal is to obtain possibly highly accurate classifiers in the subsequent transfer-learning step. Thus, it is beneficial to pre-train the base model on another public dataset which is more similar to the one used in the ultimate transfer-learning task.

### I.2  IMAGE CLASSIFIER TRANSFER-LEARNING

During the transfer-learning step we realize the following general procedure.

Table 7: The head-NN size/architecture and training details in the transfer-learning step. $FC(k)$ denotes the fully-connected layer with $k$ neurons. In each experiment the head-NNs are trained via the standard gradient descent (full batch, no momentum) with the learning rate $lr$ and weight decay $\lambda_{WD}$. The initial weights of the head-NN are drawn from $\mathcal{N}(0, \sigma_{init}^2)$ and the biases are initialized to 0. The training loss is always the cross-entropy loss. $N$ is the training set size of a given shadow model.

| Experiment | Head-NN Architecture | $\sigma_{init}$ | $lr$ | $\lambda_{WD}$ | Epochs |
|---|---|---|---|---|---|
| MNIST | Input $\to FC(10) \to$ Output | 0.002 | 0.01 | $10^{-5}$ | $26 + 3N/5$ |
| CIFAR | Input $\to FC(10) \to$ Output | 0.0002 | 0.01 | $10^{-4}$ | $38 + N$ |
| CelebA | Input $\to FC(4) \xrightarrow{ReLU} FC(1) \to$ Output | 0.0002 | 0.02 | $10^{-2}$ | 100 |

Table 8: The mean and the standard deviation of the test accuracy of the transfer-learned image classifiers. Calculated over a sample of 5000 classifiers.

|  | $N = 10$ | $N = 40$ |
|---|---|---|
| Experiment | Test Accuracy | Test Accuracy |
| MNIST | $0.812 \pm 0.041$ | $0.920 \pm 0.017$ |
| CIFAR-10 | $0.587 \pm 0.049$ | $0.794 \pm 0.022$ |

1. We remove the output neurons from each of the pre-trained neural nets described in the Subsection I.1. The outputs of the neural net obtained this way are the post-activation outputs of the penultimate layer of the original neural net.

2. For all the images coming from the datasets (MNIST, CIFAR-10 and CelebA) we pre-compute their corresponding deep features vectors. This is done by feeding the neural nets from point 1 above by the images from the corresponding dataset and saving the obtained (penultimate post-activation) outputs. The input images are transformed according to the transformations specified in the Subsection I.1. This way, we replace the image datasets with their corresponding deep-feature datasets.

3. In each of the experiments we define a new head-NN which takes the pre-computed deep features as inputs and outputs the predictions. Due to the different nature of each of the transfer-learning tasks, in each experiment the head-NN has a slightly different size/architecture. We also train the head-NNs as part of the shadow model training (both for the training shadow model sets and the validation shadow model sets), see Algorithm 1. The head-NN architecture and training details are summarized in Table 7.

In the MNIST- and CIFAR-10 experiments we use balanced training sets i.e., $N = C \times M$ with $C = 10$ being the number of classes and $M$ the number of training examples per class. In the CelebA-experiment we emulate transfer-learning with unbalanced data. There, we have $M$ examples of the positive class and $9M$ examples of the negative class. In the CelebA training loss we upweight the positive class with the weight 10. Table 5 shows the mean and the standard deviation of the test accuracy of the resulting transfer-learned image classifiers. Because in this work we are evaluating the impact of DP-SGD on the classifier accuracy, it was important to optimize classifier training to obtain reasonably good accuracies. Indeed, the accuracies obtained for transfer-learning in the CIFAR-10 experiment with the training set sizes $N = 10$ and $N = 40$ are comparable with accuracies reported in the literature, see Figure 2 in (Kolesnikov et al., 2020). Note, however, that work (Kolesnikov et al., 2020) uses data augmentation to improve transfer-learning test accuracy with small datasets, so their accuracies are generally better. We have not used data augmentation in our transfer-learning experiments to keep the setup simple. In the face recognition experiment with CelebA data it has been challenging to train good image classifiers on small datasets. The true-positive rate of the classifiers (corresponding to the correct classification of the minority class) in this experiment has varied significantly between the shadow models, see Table 9.

Table 9: The mean and the standard deviation of the classification test accuracy, classification true-positive rate ($TPR$, correct classification of the minority class) and classification true-negative rate ($TNR$, correct classification of the majority class) of the transfer-learned face-recognition classifiers trained on unbalanced data in the CelebA-experiment. Calculated over a sample of 5000 classifiers.

| $N$ | Classification Acc. | Classification $TPR$ | Classification $TNR$ |
|---|---|---|---|
| 10 | $0.914 \pm 0.037$ | $0.434 \pm 0.286$ | $0.967 \pm 0.035$ |
| 40 | $0.935 \pm 0.030$ | $0.645 \pm 0.216$ | $0.967 \pm 0.022$ |

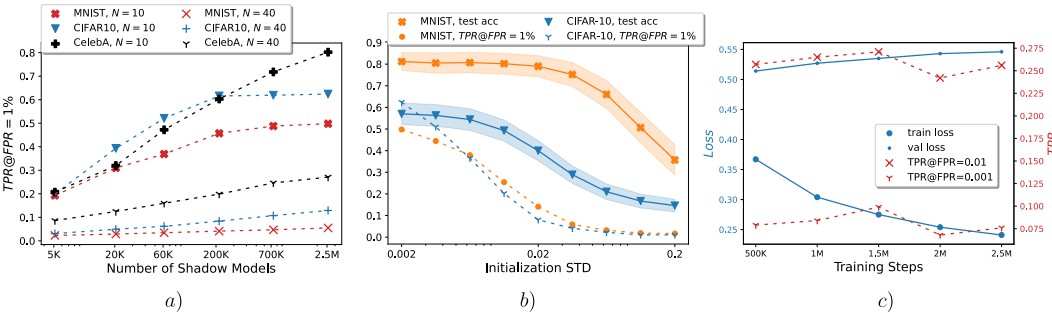

Figure 21: Factors affecting reconstruction in the white-box attack. a) Reconstruction attack $TPR$ as a function of the number of shadow models. b) Influence of weight initialization STD on reconstruction $TPR$, $FPR$ and attacked classifier accuracy for the training dataset size of $N = 10$. c) Training and validation loss and $TPR$ changes at $FPR = 0.01$ during the reconstructor NN training in the CelebA experiment with $N = 40$.

## I.3 FACTORS AFFECTING RECONSTRUCTION IN THE WHITE-BOX ATTACK

Our attack remains robust in a wide range of circumstances which we describe in detail below.

*1) Number of Shadow Models.* To obtain high reconstruction $TPR$ it is necessary to produce as many shadow models as possible given the computational resources at hand – see Fig. 21a. It is relatively easy to produce several millions of shadow models – this task is massively parallelizable and training a single shadow model is fast.

*2) Attacked Model Training Set Size.* If the number of images in the classifier's training set grows, the reconstruction $FPR$ of our attack becomes higher (see Table 2). Intuitively, this means that it becomes easier for the reconstructor to "guess" a close match to one of the target images by outputting a generic representative of the target class. We have noticed that this can be mitigated to some extent by training the reconstructor for longer, see Fig. 21c. However, training for too long may reduce $TPR$ due to overfitting, especially when there is not enough shadow models to train on or when the image dataset is small and there are many overlaps between the shadow training datasets.

*3) Conditional/Non-conditional Reconstructor NN.* For the same reasons as in point 2) above non-conditional reconstructor NNs have worse $TPR$-at-low-$FPR$. In the CIFAR experiment with the non-conditional reconstructor and $N = 10$ we got reconstruction $TPR = 31.2\%$ at $FPR = 1\%$. As with GANs, the non-conditional reconstructor is susceptible to the mode collapse - its output images tend to be generated from only one or two of the classes. Thus, conditional reconstructor NNs can retrieve much more information about the training dataset.

*4) Reconstructor conditioning mismatch.* We can successfully reconstruct training data when the reconstructor is conditioned on a different set of classes than the attacked classifier is trained to distinguish. We have attacked binary classifiers (even/odd for MNIST and animal/vehicle for CIFAR-10) with reconstructors conditioned on ten classes. We have obtained $TPR = 26.7\%$ and $TPR = 32.5\%$ at $FPR = 1\%$ for MNIST and CIFAR-10 respectively (training set size $N = 10$).

*5) Training Algorithm $\mathcal{A}$: SGD/Adam.* Results from Table 2 concern attacks on classifier models trained with $SGD$. We can also successfully attack models trained with Adam (Kingma & Ba, 2015), although with slightly lower success. We have obtained reconstruction $TPR = 34.3\%$ and

Table 10: The head-NN size/architecture and training details in the transfer-learning step in the binary classification tasks. $FC(k)$ denotes the fully-connected layer with $k$ neurons. In each experiment the head-NNs are trained via the standard gradient descent (full batch, no momentum) with the learning rate $lr$ and weight decay $\lambda_{WD}$. The initial weights of the head-NN are drawn from $\mathcal{N}(0, \sigma_{init}^2)$ and the biases are initialized to 0. The training loss is always the binary cross-entropy loss. $N$ is the training set size of a given shadow model.

| Experiment | Head-NN Architecture | $\sigma_{init}$ | $lr$ | $\lambda_{WD}$ | Epochs |
|---|---|---|---|---|---|
| MNIST | Input $\rightarrow FC(8) \xrightarrow{ReLU} FC(1) \rightarrow$ Output | 0.002 | 0.1 | $10^{-4}$ | $26 + 3N/5$ |
| CIFAR | Input $\rightarrow FC(8) \xrightarrow{ReLU} FC(1) \rightarrow$ Output | 0.0002 | 0.05 | $10^{-4}$ | $38 + N$ |

$TPR = 54.2\%$ at $FPR = 1\%$ for MNIST and CIFAR-10 respectively (training set size $N = 10$). We have not observed significant changes in reconstruction success rates when comparing full-batch SGD/Adam vs. mini-batch SGD/Adam with batch size $B = N/2$.

*6) Attacked Model Weight Initialization.* We initialize the weights of the classifier according to i.i.d. normal distribution $\mathcal{N}(0, \sigma^2)$ and initialize the biases to zero. Unless stated otherwise, we choose $\sigma = 0.002$ for the MNIST and CIFAR experiments and $\sigma = 0.0002$ for the CelebA experiments. Fig. 21b shows how the reconstruction $TPR$, $FPR$ and classifier accuracy depend on $\sigma$ for the training dataset size of $N = 10$. Higher values of $\sigma$ decrease reconstruction $TPR$ and increase $FPR$ eventually destroying our attack. However, in order to completely prevent our reconstruction attack one needs to use values of $\sigma$ that result in suboptimal classifier training and highly reduced accuracy. In contrast, reconstruction attacks in the threat model of the informed adversary require access to the exact initialized weights and fail otherwise (Balle et al., 2022).

*7) Attacked Model Underfitting/Overfitting.* We have trained the attacked models for 1, 32/48 (optimal) and 512 epochs in the MNIST/CIFAR experiments respectively with training set size $N = 10$. We have not noticed any significant differences in the corresponding reconstruction success rates.

*8) Transfer Learning with Data Augmentation.* We have trained the attacked models in MNIST and CIFAR experiments with data augmentation consisting of random rotations by $[-15°; 15°]$ and random horizontal flips (only for CIFAR). Transfer learning with data augmentation is more expensive because the deep features have to be recalculated for every training batch. This extends the shadow model training times by a factor of $\sim 10$. We have obtained reconstruction $TPR = 59.1\%$ and $TPR = 67.2\%$ at $FPR = 1\%$ for MNIST and CIFAR-10 respectively (training set size $N = 10$). Thus, data augmentation makes our reconstruction attack more effective.

*9) Out-of-distribution (OOD) Data.* CIFAR-100 is often used for OOD benchmark tests for CIFAR-10 (Fort et al., 2021). To study the effectiveness of the reconstructor NN trained on OOD data, we have trained the conditional reconstructor NN on shadow models (classifiers) trained on $N = 10$ CIFAR-100 images that were randomly assigned the class labels $c \in \{0, 1, \ldots, 9\}$. We have subsequently used such a reconstructor to attack classifiers that were trained on CIFAR-10 images (with original labels) obtaining reconstruction $TPR = 34.4\%$ at $FPR = 1\%$ and $TPR = 11.1\%$ at $FPR = 0.1\%$. This shows that our attack can be effective even with limited information about the prior training data distribution $\pi$.

Next, we provide some more details concerning the experiments described in this section. Unless stated otherwise, the hyper-parameter and architecture configuration in each experiment has been the same as described in Section I.2.

In point 4 (reconstructor conditioning mismatch) we have used the hyper-parameters and head-NN architectures for the binary image classifiers as specified in Table 10.

In point 6 (attacked model weight initialization) we have varied $\sigma_{init}$ and kept all the other hyper-parameters fixed to the values specified in Table 7. This way, we have produced the plots presented in Fig. 21b.

Table 11: The head-NN size/architecture and training details in the transfer-learning step in the one-shot 10-way classification tasks for MNIST and CIFAR-10 in the black-box attack experiments. $FC(k)$ denotes the fully-connected layer with $k$ neurons. In each experiment the head-NNs are trained via the standard gradient descent (full batch, no momentum) with the learning rate $lr$ and weight decay $\lambda_{WD}$. The initial weights of the head-NN are drawn from $\mathcal{N}(0, \sigma_{init}^2)$ and the biases are initialized to 0. The training loss is always the cross-entropy loss.

| Experiment | Head-NN Architecture | $\sigma_{init}$ | $lr$ | $\lambda_{WD}$ | Epochs |
|---|---|---|---|---|---|
| MNIST | Input $\to FC(10) \xrightarrow{ReLU} FC(10) \to$ Output | 0.01 | 0.2 | $10^{-3}$ | 100 |
| CIFAR-10 | Input $\to FC(16) \xrightarrow{ReLU} FC(10) \to$ Output | 0.007 | 0.1 | $10^{-3}$ | 100 |

## I.4 TECHNICAL DETAILS OF THE BLACK-BOX ATTACK

Our black-box attack consists of two phases.

1. We first run the hard-label weight extraction attack by Carlini et al. (2025) (see references therein for the publicly available implementation which we have used). This attack extracts hidden layers' weights which consist of 10 and 16 neurons in the MNIST and CIFAR-10 experiments respectively (see Table 11). Because this is a hard-label black-box attack (meaning that the attacker can only query inputs and the associated heard label decisions of the classifier), it is fundamentally impossible to reconstruct the weights exactly. This is because the hard-label decisions of the classifier remain invariant under transformations which permute and rescale rows of the weight-matrix of a given hidden layer (i.e. permute and rescale neurons, each neuron by a possibly different strictly positive scale factor) and simultaneously permute/rescale the columns of the weight-matrix of the subsequent layer according to the same permutation and reciprocal scale factors (Carlini et al., 2025). This symmetry is reflected in the output of the hard-label reconstruction attack; the reconstructed neurons are returned in a random order and their weight-vectors have norm one. The weights are also reconstructed up to a certain additive error which in our experiments was of the order of $\sim 10^{-9}$ which is very small.

2. The reconstructor-NN takes as input the weights that were obtained through the hard-label black-box weight extraction attack. To efficiently generate the shadow models for the training set of the reconstructor-NN we are simulating the output of the weight extraction instead of performing the weight extraction on each training-shadow model. This is done by i) rescaling each neuron from the hidden layer to unit $L_2$-norm, ii) randomly permuting the neurons of the hidden layer, iii) adding random-normal noise of mean-zero and scale $10^{-9}$ to the weights to simulate the weight extraction error. The exact (not simulated) weight extraction is performed on each validation-shadow model, however we have found that the reconstruction $TPR$ is approximately the same when calculated using the simulated weight extraction for the validation-shadow models.

Note that the hard-label weight extraction in Carlini et al. (2025) has been primarily tested on deep neural networks where it takes a long time to extract all the weights from the hidden layers (several thousands of seconds per neural net). However, the head-NNs in transfer learning are shallow, hence the timings of the weight extraction are much shorter. In the MNIST-setup from Table 11 the weight extraction took 73 seconds per shadow model.

## I.5 TIMINGS OF THE DATA RECONSTRUCTION ATTACK

Table 12 shows timings for shadow model generation and reconstructor-NN training. Shadow model generation times were obtained by training $2.56\,M$ shadow models. For shadow-model timings we have used head-NN specifications from the white-box attack (see Table 7), however the timing for the head-NNs used in the black-box attack experiments were approximately the same. For shadow model training we used $\sim 400$ CPU cores of Intel Xeon CPU E5-2680 v4 @ 2.40GHz in parallel (each such CPU has 14 cores) which is the amount compute available on almost every HPC cluster. The reconstructor-NN training times depend strongly on the input size of the reconstructor-NN. This input size is the same in the one-shot and four-shot experiments (since the head-NN architectures are

| | $t_{shadow}$ ($2.56 \times 10^6$ models) | | $t_{train}$ ($10^6$ steps) | |
|---|---|---|---|---|
| Experiment | $N = 10$ | $N = 40$ | White-Box | Black-Box |
| MNIST | $0.8\,h$ | $1.1\,h$ | $18\,h$ | $18\,h$ |
| CIFAR-10 | $1.2\,h$ | $1.8\,h$ | $41\,h$ | $81\,h$ |
| CelebA | $6.3\,h$ | $6.3\,h$ | $74\,h$ | $72\,h$ |

Table 12: Timings for shadow model generation ($t_{shadow}$, using $\sim 400$ CPU cores) and reconstructor-NN training ($t_{train}$, single GPU).

the same), but different for white-box and black-box attacks. The reconstructor-NN training times $t_{train}$ were calculated for GeForce RTX 3090 GPU.

There are several factors that contribute to the efficient generation of shadow models: i) the task is massively parallelizable, ii) the amount of training data is just a few tens of examples, iii) we train only the head-NN which contains only about $\sim 10$ neurons, iv) we do not use data augmentation, so the deep features can be pre-computed before the shadow-model training which avoids the expensive forward-prop through the complex base model during shadow model training. We have also repeated the CIFAR-10 and MNIST experiments using shadow model training with data augmentation – this has extended the above specified $t_{shadow}$-times by a factor of $\sim 10$ (due to the forward-prop through the base model of every training batch). Thus, even assuming data augmentation our attack is perfectly feasible provided that one has access to a cluster of a few tens of multi-core CPUs.

## J    OUR RECONSTRUCTION ATTACK UNDER DP-SGD

We have made sure that the DP-SGD training hyper-parameters were chosen to have as small an impact on trained image classifier's accuracy as possible. To this end, we have followed the hyper-parameter selection procedure outlined in (Ponomareva et al., 2023). To keep this paper self-contained, we summarize this procedure below. The hyper-parameters in DP-SGD are the gradient clipping norm $C$, noise multiplier $\sigma_{noise}$, poisson sampling rate $q = B/N$ ($B$ is the mini-batch size and $N$ is the size of the training set), the number of training epochs and the learning rate. We have trained the shadow models with weight decay.

1. The batch size $B$ should be as large as computationally feasible. We choose $B = N - 1$ (the largest possible allowing non-trivial mini-batch Poisson sampling). Another factor to take into account is the fact that training for too many epochs will negatively impact model's accuracy due to overfitting. In line with some transfer-learning guidelines (Chollet) we have worked with the optimal $n_E$ (number of training epochs) specified in Table 7 in the column "Epochs".

2. Tune the other hyper-parameters (learning rate and weight decay) in the non-private setting (without DP-SGD) with the chosen $B$ and $n_E$. The optimal learning rate and weight decay we have found are the same as specified in 7.

3. Choose the clipping norm $C$ by training with DP-SGD with $\sigma_{noise} = 0$ and keeping all the parameters selected in previous steps fixed. This can be done by one or more parameter swipes in the log-scale. Small values of $C$ adversely affect the accuracy of the classifier. $C$ should be chosen so that it is small, but at the same time has only slight effect on classifier's accuracy ($1 - 2$ p.p.) when compared to the accuracy of the non-private classifier, see Figure 22.

4. Compute the noise multiplier $\sigma_{noise}$ that makes the model $(\epsilon, \delta)$-DP via the privacy accountant (Abadi et al., 2016) with $C$ and $B$ found in the previous steps. In all experiments we take $\delta = N^{-1.1}$.

5. Do the final learning rate adjustment with $C$, $B$, $\sigma_{noise}$ and weight decay found in the previous steps.

By performing the above hyper-parameter selection procedure we have found that the optimal training hyper-parameters for $B = N - 1$ have been the same as the ones specified in Table 7. For simplicity,

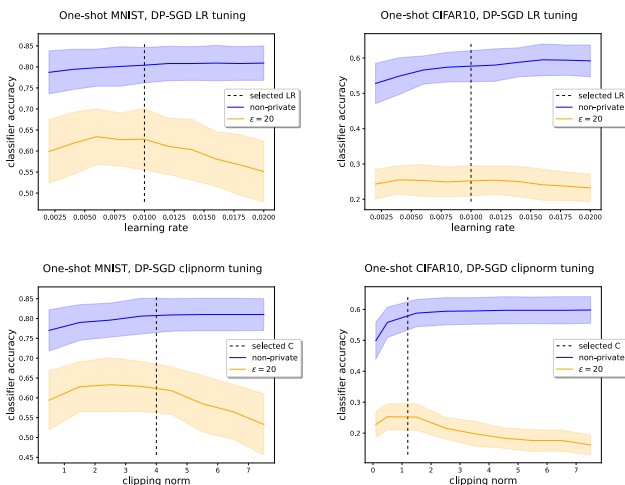

Figure 22: Plots documenting the tuning of the learning rate (top row) and the clipping norm $C$ (bottom row) in one-shot TL for MNIST and CIFAR10 for the non-private baseline and for private training with $\epsilon = 20$ which is near the "successful" defence threshold established in Section 3.2. Notice that in all the plots model utility changes very little when varying the LR and $C$ around their selected values.

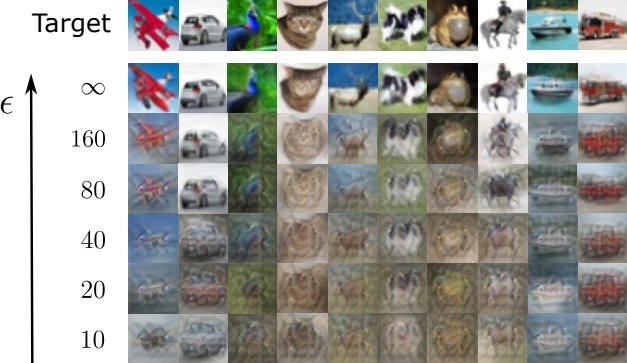

Figure 23: Reconstruction examples for white-box CIFAR-10 and $(\epsilon, \delta)$-DP models for different values of $\epsilon$. Training without DP-SGD means $\epsilon = \infty$. Classifier training set size $N = 10$.

in our experiments we have not performed the final adjustment of the learning rate. However, we have found that the learning rate values specified in Table 7 have been close to optimal in all experiments and that the model accuracy has not been very sensitive to small changes of the learning rate around these values – changing the learning rate by a factor of $\sim 5$ affects accuracy by at most $\approx 4$ p.p. for both non-private and DP-SGD runs [FIGURE TO BE ADDED]. The calibrated clipping norms $C$ were found to be $4.0$ and $1.2$ for the MNIST and CIFAR experiments respectively, see Figure 22. In all the DP-SGD experiments we use the same weight initialization as specified in Table 7. For DP-SGD training we have used the *Opacus* library (Yousefpour et al., 2021).

Figure 23 shows some further examples of reconstructions for models trained with DP-SGD.

