# OpenReview forum: "Securing Transfer-Learned Networks with Reverse Homomorphic Encryption"
_ICLR.cc/2026/Conference — Submitted to ICLR 2026_

### Official Review · Reviewer_YfZy · 2025-10-27

**Soundness:** 2
**Presentation:** 2
**Contribution:** 2
**Rating:** 4
**Confidence:** 3

**Summary:**

* This paper makes two primary contributions. First, it identifies and demonstrates a significant vulnerability in few-shot transfer learning (TL). The authors introduce new white-box and black-box training-data reconstruction attacks and show empirically that Differentially Private Stochastic Gradient Descent (DP-SGD) fails to provide an acceptable privacy-utility tradeoff in this regime. Specifically, they show that for small, few-shot datasets (e.g., $N=10$ or $N=40$ samples), defending against their attacks with DP-SGD incurs a severe classifier accuracy drop of.
* Second, to address this gap, the paper proposes a defense mechanism named "Reverse Homomorphic Encryption" (RHE). This approach is tailored for the TL use case where the training data is sensitive, but the inference data is not. RHE involves encrypting only the weights and biases of the transfer-learned "head" neural network, while leaving the large base model and the inference inputs unencrypted. The authors claim this method is computationally efficient, as it only applies HE to a shallow network (at most 3 layers) and uses a square activation function ($x^2$) to avoid costly bootstrapping. They claim this defense completely blocks training-data reconstruction attacks (both white-box and black-box) with minimal degradation in model accuracy.

**Strengths:**

* The paper's biggest strength is its extensive and rigorous supplementary material. The appendix provides a detailed and significant amount of ablations, experiments, and proofs covering many aspects of the white-box and black-box attacks, an overview of previous methods for both attacks and homomorphic encryption, and the supporting theoretical proofs.
* The motivation is compelling. It highlights a critical, practical, and under-appreciated gap in privacy-preserving ML: the failure of DP-SGD to protect few-shot transfer-learned models. The authors correctly identify that this scenario is common in sensitive domains, which are frequent applications for TL.
* The paper provides a valuable and novel contribution by designing and demonstrating sophisticated reconstruction attacks (both white-box and black-box) under a realistic "weak adversary" threat model. The empirical results in Figure 2, which plot the sharp trade-off between privacy ($\epsilon$) and utility (accuracy) when defending against these attacks with DP-SGD, are convincing.

**Weaknesses:**

* The HE component's of this work, however, feel somewhat disconnected from the core contribution, as if it were an add-on rather than a deeply integrated part of the workflow. While portions of the solution are pragmatic, its engagement with recent advancements in the field appears limited.
    * The strategy to replace ReLU with a square activation ($f(x)=x^{2}$) is a point of concern. While this avoids the cost of bootstrapping, it presents a trade-off. The validation (Table 4) is a direct comparison between $x^2$ and ReLU for shallow networks. This could be strengthened by exploring other HE-friendly activation strategies from the literature  (e.g., binary-tree ReLU implementation in CKKS [1], a fuller investigation on Chebyshev polynomials [2], etc.), which often provide a better accuracy-depth trade-off.
    * The justification for avoiding bootstrapping (due to cost) seems to be based on older benchmarks. Could the authors comment on why more recent, efficient bootstrapping methods (which can handle exact ReLU) were not considered as an alternative to replacing the activation function? (e.g., with TFHE programmable bootstrap [3] or somewhat new CKKS functional boostrapping [4]).
    * The claim of "no degradation" 3 is supported for the 1- and 2-layer heads, but the $x^2$ activation is known to introduce accuracy challenges in deeper networks. This raises questions about scalability. A study on how RHE's accuracy and performance are impacted by more complex, deeper head-NNs would significantly strengthen the paper's claims of practical utility, similar to [5].

* The performance evaluation in Table 3 would be more impactful if contextualized with other privacy-preserving frameworks.
    * Currently, the table only shows the overhead of RHE against its own unencrypted baseline. This makes it difficult to interpret the practical efficiency of the proposal relative to the state-of-the-art in private inference.
    * For instance, while the authors correctly note in Appendix A.2 that the use case for HETAL [6] is different, including its performance as a point of reference would help the reader gauge the computational cost of RHE. Could the authors provide such a comparison or benchmark against other relevant Privacy-Preserving Inference (PPI) frameworks?

* The paper's motivation hinges on attacks in the few-shot regime, which are demonstrated on dataset sizes of $N=10$ and $N=40$.
    * It would be helpful for the reader to understand how representative these small $N$ values are of real-world, sensitive transfer learning applications.
    * The authors rightly note in the limitations (Appendix B) that the attack's effectiveness decreases as $N$ increases. This prompts a key question: at what approximate dataset size $N$ does this specific attack vector become unreliable, and DP-SGD (which the paper critiques) become the more practical defense?

* Section 3, which details the novel attack, is quite dense. It synthesizes several complex components, including shadow models (Algorithm 1), a bespoke reconstructor NN architecture, the softmin loss function , and black-box weight extraction. The paper's clarity would be significantly enhanced by including a high-level diagram or summary box that illustrates the end-to-end pipeline of the attack.

* A minor point on terminology: the name "Reverse Homomorphic Encryption" may be somewhat confusing for readers (especially from the cryptography community). The innovation appears to be in the application (encrypting weights rather than inputs) rather than a new cryptographic "reverse" primitive. This is a clever form of partial model encryption, which has been previously explored [7], and perhaps a more descriptive name ("Weight-Space HE" or similar) might aid clarity.

References: \
[1] Eunsang Lee et al., ”Low-complexity deep convolutional neural networks on fully homomorphic encryption using multiplexed parallel convolutions”. *International Conference on Machine Learning*, 2022. \
[2] L. Rovida and A. Leporati, “Encrypted Image Classification with Low Memory Footprint Using Fully Homomorphic Encryption”. *International Journal of Neural Systems*. 2024 \
[3] Ilaria Chillotti, Marc Joye, and Pascal Paillier. “Programmable bootstrapping enables efficient homomorphic inference of deep neural networks.” *Cyber Security Cryptography and Machine Learning*, 2021. \
[4] Alexandru, A., Kim, A., Polyakov, Y. "General Functional Bootstrapping Using CKKS". *Advances in Cryptology*, 2025 \
[5] Njungle, Nges Brian, and Michel A. Kinsy. "Activate me!: Designing efficient activation functions for privacy-preserving machine learning with fully homomorphic encryption." *International Conference on Cryptology in Africa*. 2025. \
[6] Lee, Seewoo, et al. "HETAL: Efficient privacy-preserving transfer learning with homomorphic encryption." *International Conference on Machine Learning*. 2023 \
[7] Dongwoo Kim and Cyril Guyot. “Optimized privacy-preserving cnn inference with fully homomorphic encryption”. *IEEE Transactions on Information Forensics and Security*, 2023

**Questions:**

See weaknesses.

---

> ### Author Response · Authors · 2025-11-19
>
> We thank the reviewer for the time spent reviewing our paper and for their remarks. Below are our responses.
>
> 1. ***"The HE part is insufficiently tied to the core part of the paper. Limited engagement with recent HE advances."***
>
> Our intention is to tailor RHE to the attack setting: few-shot TL with a frozen base model and a shallow trainable head, where only the head-NN weights leak info about training examples. There, encrypting only the head weights is what prevents our reconstruction attack (and weight-based/black-box attacks), while encrypting deep feature vectors or entire models would be unnecessary and substantially more expensive.
>
> In the revision we will:
>
> * Add a bridging subsection emphasizing the motivation for RHE stemming from the security gaps in few-shot TL where we found no alternative way to prevent reconstruction.
> * Expand the related-work appendix to position RHE among recent HE-based methods, including [1–7], and clarify that our goal is to demonstrate a novel, practical, easily deployable defence for a specific threat model, not to improve the encryption algorithms.
>
> 2. ***"Square activations are not suitable for deep NNs. Other HE-friendly activations and recent bootstrapping were not fully explored."***
>
> **Our design choices prioritise deployability in common ML workflows.** In the shallow-head regime we target, we empirically show that square activations match ReLU accuracy, and they are simple to implement in CKKS and widely available in libraries such as TenSEAL.
>
> We agree that for deep networks more advanced HE-friendly activations and modern bootstrapping offer better accuracy–depth trade-offs. However, these techniques are not yet supported in the standard libraries we use. Adopting them would require switching to more specialised HE frameworks and extra engineering, which limits practical adoption by ML practitioners.
>
> We view RHE as agnostic to the specific HE schemes: more advanced schemes can be plugged into our framework for deeper heads, which we see as important follow-up work.
>
> In the revision we will:
>
> * Explicitly bound our “no degradation” claim to shallow heads.
> * Expand the discussion of alternative HE-friendly activations and recent bootstrapping methods, and justify our choice of a simple, widely supported pipeline.
>
> 3. ***"Need evidence on how RHE behaves for deeper head networks."***
>
> RHE is **not proposed as a universal solution for deep encrypted networks**. Our aim is to identify a regime where it is both necessary and efficient: few-shot TL with shallow heads, where DP-SGD cannot add enough noise without destroying accuracy.
>
> The cost of encrypting deeper models with HE has been studied extensively. We have added Figure 6 that collects per-prediction latencies vs. depth from prior CKKS-based works. This gives a realistic proxy for how RHE latency scales with depth and supports our choice to focus on shallow heads.
>
> 4. ***"Lack of context vs. state-of-the-art PPI in Table 3."***
>
> Table 3 is intended to demonstrate that our defence is feasible using a popular library (TenSEAL), not to compete directly with general-purpose PPI frameworks. Direct runtime comparison is difficult because frameworks such as HETAL encrypt inputs and deep features and aim at end-to-end private inference, whereas we encrypt only a small TL head during training. We have added the latency-versus-depth figure as discussed above.
>
> 5. ***"How realistic are the small $N$ in practice, and at what $N$ does the attack become ineffective so that DP-SGD is preferable?"***
>
> Few-shot and even one-shot TL are common in sensitive domains where labeled data are scarce: e.g., medical imaging, network intrusion detection using sensitive logs, biometrics, and financial fraud detection. We have added references to support this point.
>
> There is no universal threshold at which DP-SGD becomes preferable. The crossover depends on i) feature dimension and data distribution (cf. our MNIST vs. CIFAR-10 results), ii) reconstructor capacity and adversary compute resources, and iii) the acceptable DP parameters $(\epsilon,\delta)$. Our experiments already show that as $N$ grows, attack quality decreases while DP-SGD utility improves. For moderate/large $N$ DP-SGD becomes more attractive and RHE is no longer necessary, although still competitive when only a shallow head has been trained.
>
> 6. ***"Section 3 is dense; the attack is hard to follow".***
>
> Added attack flowcharts in Figure 2. Our repo [https://anonymous.4open.science/r/training-data-reconstruction-1EB2](https://anonymous.4open.science/r/training-data-reconstruction-1EB2) already provides detailed instructions for reproducing the attack.
>
> 7. ***“Reverse Homomorphic Encryption” suggests new cryptographic algorithm***
>
> We clarify that RHE does not propose a new encryption algorithm and the novelty lies in reversing the usual role of ciphertexts. We have retitled the paper accordingly.

---

> ### Comment · Reviewer_YfZy · 2025-11-26
>
> I thank the reviewers for their detailed response to my questions and responsive updates to the manuscript. I also particularly like the responses to the important questions that HBNF brought up. Because of these timely and detailed responses which address my original issues with the manuscript, I have decided to raise my score. Best of luck!
>
> Soundness: 2 $\rightarrow$ 3 \
> Presentation: 2 $\rightarrow$ 3 \
> Overall score: 4 $\rightarrow$ 6

---

> > ### Author Response · Authors · 2025-11-28
> >
> > Thank you for the efforts spent on reviewing our paper and for the insightful questions. We are glad our responses were useful!

---

### Official Review · Reviewer_dMgT · 2025-10-31

**Soundness:** 3
**Presentation:** 2
**Contribution:** 2
**Rating:** 4
**Confidence:** 4

**Summary:**

This paper experimentally demonstrates that existing differential privacy defense methods are ineffective in small-sample transfer learning. It also proposes a defense mechanism based on Reverse Homomorphic Encryption (RHE), which prevents data reconstruction attacks without degrading classifier performance.

**Strengths:**

1. This paper proposes effective white-box and black-box reconstruction attacks, which demonstrate stronger performance under realistic threat models compared to existing methods.
2. The experiments validate the applicability of the proposed attack methods on different datasets, showcasing their effectiveness and robustness against existing defense methods, such as DP-SGD.
3. The paper introduces the Reverse Homomorphic Encryption (RHE) mechanism, which effectively defends against data reconstruction attacks while providing high security without significantly degrading classifier performance.

**Weaknesses:**

1. The paper primarily focuses on the few-shot transfer learning scenario, but both the abstract and introduction fail to provide a systematic description of the research background.
2. The authors propose a novel reconstruction attack method, but there is a lack of detailed steps for the specific attack process. It appears to be a simple combination of several existing works, lacking novelty.
3. The paper proposes using Reverse Homomorphic Encryption (RHE) to defend against reconstruction attacks, but it assumes that only the head neural network is fine-tuned during the transfer learning process. There is no discussion on the generalizability of this assumption in practical applications.

**Questions:**

1. Provide a more systematic description of the research background. Does the proposed approach generalize to non-transfer learning or non–few-shot transfer learning scenarios?
2. Provide a detailed description of the proposed reconstruction attack process, emphasizing the innovative aspects.
3. Since the method assumes that only the head neural network is fine-tuned during transfer learning, clarify whether this assumption holds broadly in practical applications. How does the proposed approach perform compared to fine-tuning the entire model?

---

> ### Author Response · Authors · 2025-11-19
>
> We thank the reviewer for the helpful comments and address each point below.
>
> Q1.1 **“Provide a more systematic description of the research background.”**
>
>    We agree the background can be more systematic. Our paper targets TL on small private datasets, where standard defenses such as DP-SGD often incur severe utility loss. In the revised Introduction we explain that few-shot TL is common in practice and will i) add a structured TL background and ii) map the RHE-relevant threat families, including beyond TL.
>
> Q1.2 **“[…] generalization to non–few-shot TL scenarios and beyond TL?”**
>
>    RHE is not limited to few-shot TL. It protects any setting where the adversary relies on plaintext weights or model decisions, including model-stealing, inversion/reconstruction, and membership-inference attacks. It also applies to non–few-shot TL and full-model fine-tuning (with increasing cost for deeper or entire models). A key contribution is identifying few-shot TL as especially suitable for efficient HE-based defense, since there typically only the shallow head NN is trained. The cost of encrypting deeper models is well studied in the HE literature. We have added Figure 6 summarizing how HE per-prediction latency scales with NN depth for full-network CKKS encryption, supporting our choice to focus on shallow heads. We will also briefly discuss other compatible threat families such as MIA, model inversion, and model stealing (expanding Section 4).
>
> Q2 **“Provide a detailed description of the reconstruction attack process, emphasizing the novelty.”**
>
>    We have added a flowchart (Figure 2) summarizing the attack. Our code repo provides detailed, step-by-step instructions:
>    [https://anonymous.4open.science/r/training-data-reconstruction-1EB2](https://anonymous.4open.science/r/training-data-reconstruction-1EB2)
>
>    Our attack is substantially harder than standard inversion demonstrations, especially in TL and under a weak adversary:
>
>    * **No gradients and minimal side information.** The attacker only sees the final fine-tuned model (white-box) or hard-label outputs (black-box). Prior work typically assumes access to intermediate gradients, logits, or all-but-one training examples. Instance-level recovery from only end-state weights or hard-label decisions has not been studied before.
>    * **Robust to training-pipeline uncertainty.** Unlike prior works, our attack remains effective without knowing initial weights or mini-batch seeds and in the presence of data augmentation in the TL dataset.
>    * **Hard-label black-box reconstruction.** We are the first to realise the hardest black-box setting where the attacker observes only hard-label outputs.
>    * **Instance-level, not prototype-level, recovery.** Many prior “reconstruction” methods yield class prototypes or nearest neighbors. Our attack aims to reconstruct specific training instances, handling data permutation symmetries and ambiguities while still producing semantically faithful images.
>    * **Novel, rigorous evaluation.** We use ROC and TPR-at-low-FPR based on LPIPS with appropriate thresholds, instead of naive metrics (MSE, cherry-picked examples), giving stricter, more informative evaluation for practitioners.
>
>    This robustness comes from shifting to single-instance reconstruction: for each class, the reconstructor NN discovers the data point most prone to reconstruction. To realize this, we i) introduce a softmin-based loss that automatically selects the most vulnerable instance among $N$ candidates (and solves key data permutation ambiguities), and ii) design a reconstructor architecture that scales to higher-resolution data and reconstructions beyond the common $32\times 32$ setting.
>
> Q3 **“The paper assumes that only the head neural network is fine-tuned during TL. Does this assumption hold broadly in practice? How about fine-tuning the entire model?”**
>
>    * Head-only fine-tuning is widely used in small-data regimes. We show that RHE is most efficient when encrypting only a few top layers, suggesting that practitioners with sensitive data should favor partial fine-tuning to preserve per-prediction efficiency. Such TL methods are well established and can match full-model fine-tuning accuracy when the base model is strong (see references in the preliminarily revised paper). The number of encrypted layers that is practical depends on computational resources and required throughput; full-model fine-tuning is more common when private datasets are larger and domain shift is substantial.
>    * RHE still protects the data when more layers are fine-tuned and encrypted, with inference latency depending on the size of the encrypted portion (see Fig. 6 in the revision).
>
> **Edits**
>
> 1. Added an attack flowchart and sharpened the presentation of attack novelty.
> 2. Added Fig. 6 (HE latency vs. NN depth).
> 3. Explained that head-only tuning is practical and widely applicable.
> 4. Will add a structured TL background and map the RHE-relevant threat families.

---

> > ### Author Response · Authors · 2025-11-28
> >
> > Thank you again for the time and effort you have put into reviewing our submission. We have responded to all comments and posted our detailed rebuttal. Please look over our responses and consider whether they address your concerns. Your timely response will help us address any remaining issues before the end of discussions window.
> >
> > We appreciate your feedback and thoughtful evaluation of our work.

---

### Official Review · Reviewer_HBNF · 2025-10-31

**Soundness:** 3
**Presentation:** 3
**Contribution:** 2
**Rating:** 6
**Confidence:** 3

**Summary:**

This paper investigates the vulnerability of transfer-learned neural networks to training-data reconstruction attacks and proposes a novel encryption-based defense mechanism named Reverse Homomorphic Encryption (RHE). The authors first demonstrate that differentially private stochastic gradient descent (DP-SGD), the de facto standard for privacy-preserving training, fails to protect few-shot transfer learning (TL) models without severely degrading accuracy. They design sophisticated white-box and black-box reconstruction attacks that succeed under a realistic “weak adversary” threat model, recovering sensitive training data from TL classifiers trained on small per-class datasets. To counter these threats, the authors introduce RHE—a role-reversed form of homomorphic encryption that encrypts the transfer-learned weights rather than the input data, ensuring that only a trusted party with a private key can access decrypted outputs. Experiments on MNIST, CIFAR-10, and CelebA demonstrate that RHE blocks all reconstruction and inference attacks while maintaining model utility and providing efficient inference on commodity hardware.

**Strengths:**

1.  They introduce new and effective reconstruction attacks (both white-box and a novel hard-label black-box) specifically tailored to a realistic adversary model in the few-shot transfer learning setting.
2.  The authors show that the de facto defense, DP-SGD, fails to mitigate these attacks without incurring a severe, unacceptable loss in classifier utility.
3.  The paper proposes a highly effective defense mechanism: Reverse Homomorphic Encryption (RHE). The properties of this defense are particularly impressive, as the authors claim it provides a complete defense against a wide range of attacks (reconstruction, MIA, and property-inference). The authors show it incurs no degradation in classifier utility. Furthermore, they demonstrate that this HE-based approach is computationally practical, even without high-performance hardware.
4.  The paper's claims are further strengthened by a principled, Neyman-Pearson-based scheme for evaluating reconstruction attacks. By rigorously accounting for both false positives and false negatives, this evaluation framework adds a layer of robustness and credibility to the attack results presented.

**Weaknesses:**

Please refer to my questions below.

**Questions:**

1\. Questions Regarding the DP-SGD Baseline Comparison
======================================================

I note that the paper draws a very strong conclusion in Section 3.2 regarding the catastrophic utility loss of DP-SGD (e.g., >30 p.p. accuracy drop), which serves as a primary motivation for the RHE solution.

My questions are focused on the setup of this baseline comparison:

   It is well-established that the utility of DP-SGD is exceptionally sensitive to its hyperparameter configuration, particularly the gradient clipping norm ($C$) and the noise multiplier.
   The authors state that an "optimal $C$" was used, but the manuscript does not appear to present the detailed tuning experiments, ablation studies, or sensitivity analyses (e.g., plotting accuracy vs. $\\epsilon$ for different $C$ values) that would be needed to substantiate this claim of optimality.
   More critically, the authors explicitly state in Appendix J that, "For simplicity, in our experiments we have not performed the final adjustment of the learning rate".

I am left wondering whether the performance reported is truly the strongest baseline (best achievable utility) for DP-SGD under this privacy budget, or if it might reflect a sub-optimal parameter choice. Clarifying this is crucial for a fair assessment of RHE's comparative advantage. Could the authors please provide more experimental details to support this argument?

2\. Questions Regarding the Generalizability to Non-Frozen Paradigms
====================================================================

The proposed black-box attack and the RHE defense both appear to ingeniously exploit a specific property of the "frozen backbone" transfer learning paradigm: that the base model remains static and all sensitive information is isolated within the Head-NN.

This raises significant questions about the generalizability of these methods to other widely-used, modern fine-tuning scenarios:

1.  Regarding the Black-Box Attack: The attack's success (Section 3.1) hinges on the adversary's ability to locally compute the precise Head-NN input features ($f(X)$) using an identical, publicly available base model. However, if a victim employs Full Model Fine-tuning or Parameter-Efficient Fine-Tuning (PEFT) techniques (e.g., LoRA), the base model's parameters are modified by the sensitive data. Would this not break the identity between the victim's and attacker's base models, thus invalidating the attacker's local computation of $f(X)$? Could the authors comment on whether the entire black-box attack chain would fundamentally fail in such scenarios?

2.  Regarding the RHE Defense: The security model of RHE appears to be based entirely on encrypting the Head-NN. In PEFT paradigms, however, the influence of sensitive data "leaks" into and is stored within unencrypted components of the base model (e.g., the LoRA weights). Would the security guarantees of RHE still hold in this case?


Could the authors please clarify whether the proposed attack and defense methodologies are inherently limited to the "frozen backbone" transfer learning paradigm? Or, do they (and if so, how) have the potential to be generalized to these other, more mainstream fine-tuning techniques?

---

> ### Author Response · Authors · 2025-11-19
>
> We thank the reviewer for the thoughtful and constructive feedback. Below we respond point-by-point and will revise the paper accordingly.
>
> Q1. ***DP-SGD baseline and hyperparameter tuning***
>
> We agree that DP-SGD utility is highly sensitive to the clipping norm $C$ and learning rate (LR), and that this must be carefully documented. In all DP-SGD experiments we followed the “How to DP-fy ML” guidelines (Appendix J):
>
> * First establish a non-private baseline.
> * Sweep $C$ and select the smallest value such that, without noise, accuracy is within 1–2 p.p. of the non-private baseline.
> * Note that larger $C$ at a fixed privacy budget does not improve accuracy; it only weakens privacy.
>
> In the revision we will i) replace “optimal $C$” with “calibrated $C$”, and ii) add representative accuracy-vs-$C$ plots to make this calibration transparent.
>
> As stated in Appendix J, we did not perform a final fine-grained LR sweep. In our few-shot transfer-learning regime, the accuracy landscape around the chosen LR values (Table 10) is essentially flat: changing the LR by a factor of $\sim 5$ affects accuracy by at most $\approx 4$ p.p. for both non-private and DP-SGD runs.
>
> Thus, additional LR tuning cannot close the $\approx 30$ p.p. gap at the target privacy budget. We will add accuracy-vs-LR plots to Appendix J to document this flatness and clarify that **the reported DP-SGD curves correspond to a reasonably tuned configuration rather than a weak baseline**.
>
> Q2. ***Generalizability beyond frozen-backbone transfer learning***
>
> Our work focuses on the widely used “frozen backbone + trained head” setup, stated explicitly at the beginning of Section 4.1.
>
> Q2.1 ***Black-box attack under full/PEFT fine-tuning***
>
> If the victim uses full-model fine-tuning or PEFT (e.g., LoRA), both backbone and head are updated on sensitive data. The public backbone then no longer matches the fine-tuned backbone, and our attack chain becomes substantially more difficult (although in principle realisable with sufficient resources).
>
> The black-box weight-extraction attack of Carlini et al., 2025 applies to fully connected NNs of arbitrary depth (with extraction cost growing with depth). If the backbone were fully connected, one could extract its weights and feed them to the reconstructor NN. Typical backbones are not fully connected (they mix convolution, attention, batch norm, etc.), and we are not aware of hard-label black-box weight extraction for such general architectures. However, this is a quickly evolving field.
>
> A further difficulty is the need to train millions of shadow models to build the reconstructor’s training set. With full-model fine-tuning, each shadow model is an expensive fine-tune requiring access to tens of high-end GPUs, significantly raising the attack cost. Such resources might be available to a sufficiently powerful adversary.
>
> Q2.1 ***RHE under full/PEFT fine-tuning***
>
> In PEFT (e.g., LoRA) or full fine-tuning, sensitive information is stored in the modified backbone parameters, which are not encrypted in our current design. Private information can leak from these unencrypted components (e.g., via reconstruction or inversion), and RHE as proposed here does not prevent this.
>
> To address this in the full fine-tuning case, one would need to additionally encrypt the entire backbone. Encrypting more layers (or the whole model) significantly increases the per-prediction latency. A key contribution of our work is to identify few-shot TL as particularly suitable for efficient HE-based defense: only the shallow head NN is trained on the small TL dataset, so encrypting just the head is cheap. The cost of encrypting deeper models has been extensively studied in the HE literature. In the revised paper we have added Figure 6 summarizing how HE per-prediction latency grows with NN depth, based on our literature review.
>
> We therefore do not claim that the current RHE scheme fully protects settings where the backbone is fine-tuned. Generalizing RHE to these cases would require encrypting additional components or redesigning the protocol, which we view as important future work.
>
> Head-only (and partial) fine-tuning are widely used in small-data regimes, while full-model fine-tuning is more common with larger private datasets or strong domain shift. Our results show that RHE is most efficient when applied to only a few top layers, suggesting that, when training with sensitive data, practitioners should prefer partial fine-tuning to preserve per-prediction efficiency. Such techniques are well established and can match or closely approach full-model fine-tuning accuracy when the base model is strong, see the references in our resbmitted initial revision. The efficiency cost of fine-tuning and encrypting more layers under significant domain shift depends on available compute and required throughput: lower throughput allows higher per-prediction latency.

---

> ### Author Response · Authors · 2025-11-22
>
> **Follow-up response to Question 1 (parameter tuning in DP-SGD)**
>
> In the revised manuscript we have now included Figure 22 (Appendix J). This figure shows plots documenting the tuning of the learning rate (top row) and of the clipping norm $C$ (bottom row) in one-shot TL for MNIST and CIFAR10 for the non-private baseline and for private training with $\epsilon=20$ which is near the ``successful" defence threshold established in Section 3.2.
>
> One can see that in all the plots:
>
> * Model utility is near-optimal at the selected values of $C$ and LR (for MNIST this could be potentially improved by about $1$ p.p.).
> * Model utility changes very little when varying the LR and $C$ around their selected values.
>
> We have also run additional experiments where we have increased the training lengths by the factor of $\times 2$ and $\times 4$ (keeping the same $\epsilon$ and re-calibrating $C$ and $LR$) and have observed **no improvement to the reported model accuracy**.
>
> Please let us know if you have any further concerns about the DP-SGD calibration process or any other aspects of the paper.

---

> > ### Author Response · Authors · 2025-11-28
> >
> > Thank you again for the time and effort you have put into reviewing our submission. We have responded to all comments and posted our detailed rebuttal. Please look over our responses and consider whether they address your concerns. Your timely response will help us address any remaining issues before the end of discussions window.
> >
> > We appreciate your feedback and thoughtful evaluation of our work.

---

### Official Review · Reviewer_KKYk · 2025-11-01

**Soundness:** 1
**Presentation:** 3
**Contribution:** 2
**Rating:** 4
**Confidence:** 4

**Summary:**

The paper studies training-data reconstruction for few-shot transfer learning (TL). It builds strong white-box and hard-label black-box reconstruction attacks, shows that defending with DP-SGD in the few-shot regime requires very noisy settings that hurt accuracy, and then proposes “Reverse Homomorphic Encryption” (RHE): keep inputs and the frozen base model in the clear, but homomorphically encrypt the TL head’s weights (and logits), claiming this blocks reconstruction/MIA/property-inference while remaining practical for shallow heads.

**Strengths:**

Attack contribution & evaluation. New hard-label black-box path (weight extraction → reconstructor) and a cleaner Neyman–Pearson ROC criterion for quantifying reconstruction at low FPR; results show sizable TPR at 1% FPR in few-shot TL.

Pragmatic defense idea. RHE is conceptually simple (encrypt the TL head, not the inputs), and the prototype reports per-prediction latency that’s in the tens to hundreds of ms for small heads—reasonable for many TL scenarios.

**Weaknesses:**

Reading the title "Securing Transfer-Learned Networks with Reverse Homomorphic Encryption", I assume the paper is about a defense. However, most pages develop and measure the attack and the DP-SGD tradeoff. RHE is introduced later with a design sketch, implementation notes (CKKS/TenSEAL), and timing/accuracy tables, but there is no head-to-head evaluation demonstrating the attack failing under RHE (the defense claim is largely by construction: encrypt weights/logits ⇒ attacks can’t run). The defense section is comparatively brief (Sections 4–4.2) and focuses on mechanics/latency rather than a formal security proof or empirical “we tried our own attack and it cannot proceed.”

Based on this weakness, I cannot say a sound experimental support for your claimed contribution 4, 5.

Meanwhile, I think the paper concentrate to a trivial problem, the reconstruction threats. I think the author should reorganize the paper and focus on the RHE. Cause HE can not only protect the data from reconstruction attacks, but some other attacks, like model stealing attack, or can be used to watermark the model, etc. Why you focus on only one specific attack but not your protection? For me, it's like missing the forest for the trees. I suggest the author reorganize the paper following the title - focus on the protection and validate the protection. There are many content in appendix should be in the main pages.

**Questions:**

Please check the weaknesses.

---

> ### Author Response · Authors · 2025-11-19
>
> We thank the reviewer for the comments and for pointing out additional scenarios where RHE is applicable. Below we respond point by point and indicate planned revisions, and we clarify the misunderstanding concerning RHE defence proof.
>
> 1. **Scope and framing (“the paper is about HE defense, but most pages develop the attack and DP-SGD tradeoff”)**
>
> Our structure follows a standard security-paper pattern: i) define and validate a concrete threat, ii) derive defense requirements, iii) design and evaluate a deployable defense. Sections 2–3 are thus essential as they show that training-data reconstruction in few-shot TL is a real risk and that DP-SGD does not mitigate it without large accuracy loss. This motivates stronger protection for few-shot TL (common in medical imaging and other high-stake areas), where an effective defense must prevent any plaintext access to head-NN weights, logits, or decisions. RHE is designed to meet exactly this requirement. We also introduce a rigorous reconstruction evaluation, to strengthen the privacy risk assessment motivating RHE.
>
> We will add a bridging subsection motivating RHE in few-shot TL and beyond. We have retitled the paper accordingly.
>
> 2. **Lacking “head-to-head evaluation” of the attack under RHE. “The defence claim needs a formal proof.”**
>
> We believe there is a misunderstanding. **Running our reconstruction attack under RHE is ill-posed.** All known reconstruction methods require some knowledge of head-NN weights, logits, or decisions. Under RHE, none of these are available in plaintext. The meaningful checks for RHE are therefore: i) verify that an adversary never obtains plaintext weights or decisions, ii) measure accuracy loss vs. the unencrypted model, and iii) measure per-prediction latency.
>
> The paper already includes:
> * A precise API and trust model (who holds keys, what runs where, which interfaces return ciphertexts).
> * An adversary-reduction argument: any adversary that reconstructs data under RHE must effectively break the HE scheme, which has been extensively scrutinised and is costed at $\approx 2^{128}$ operations, far beyond foreseeable classical or quantum compute.
>
> We will highlight this more clearly in the Introduction. The defence claim cannot be validated by running the attack “under RHE”, nor is such an experiment required.
>
> 3. **“Focus only on reconstruction is trivial”**
>
> Realising our attack also proves the vulnerability to membership- and property-inference attacks. Our attack is substantially harder than standard inversion demonstrations, especially in TL and under a weak adversary:
>
>    * **Minimal side information.** The attacker only sees the final fine-tuned model (white-box) or hard-label outputs (black-box). Prior work typically assumes access to intermediate gradients, logits, or all-but-one training examples. Instance-level recovery from only end-state weights or hard-label decisions has not been studied before.
>    * **Robust to training uncertainty.** Unlike prior works, out attack works without the knowledge of initial weights or other random seeds and under training data augmentation.
>    * **Hard-label black-box reconstruction.** We are the first to realise the hardest black-box setting with only hard labels.
>    * **Instance-level recovery.** We target specific training instances, not just prototypes or nearest neighbours (like many prior “reconstruction” methods do).
>    * **Rigorous evaluation.** ROC and TPR-at-low-FPR based on LPIPS, rather than MSE or cherry-picked examples, giving stricter, more informative evaluation for practitioners.
>
> To achieve such robustness, we shift to single-instance reconstruction: for each class, the reconstructor finds the data-point most prone to recovery. To this end, we i) introduce a softmin-based loss that selects the most vulnerable instance among (N) candidates and ii) design a reconstructor that scales to higher-resolution inputs beyond the common  $32\times 32$. We will emphasise this novelty more clearly.
>
> 4. **“Reorganize to focus on RHE / other threats instead of reconstruction”**
>
> In few-shot TL, reconstruction currently lacks any utility-preserving defense: DP-SGD needs large noise and severely degrades accuracy. Thus, we focus on this concrete high-impact threat and then show that RHE provides a deployable protection at acceptable cost.
>
> RHE also applies to:
> * Membership and property inference,
> * Model stealing via weight extraction or black-box queries,
> * Model watermarking.
>
> Analysing all these in depth would either turn the paper into a survey or dilute the main case. We will add a discussion section mapping these additional consequences of encrypting weights/decisions in RHE.
>
> 5. **Moving appendix material to main text**
>
> We placed proofs, experimental/ablation details, and timing tables in the Appendix to stay within the 10-page limit. We will use the extra page in the camera-ready version to move selected RHE details into the main body for better readability.

---

> > ### Comment · Reviewer_KKYk · 2025-11-27
> >
> > 1. The paper reads less like a general “privacy method” paper and more like a threats-first paper: design a stronger attack, then sketch a defense. That is a valid contribution, but the title/positioning is confusing given this emphasis. Consider retitling or clarifying in the abstract that the primary contribution is threat discovery and evaluation, not a production-ready privacy mechanism. (Contrast with DP-style works that lead with formal guarantees. [1,2])
> >
> > [1]Abadi, M., Chu, A., Goodfellow, I., McMahan, H. B., Mironov, I., Talwar, K., & Zhang, L. (2016, October). Deep learning with differential privacy. In Proceedings of the 2016 ACM SIGSAC conference on computer and communications security (pp. 308-318).
> > [2]Arachchige, P. C. M., Bertok, P., Khalil, I., Liu, D., Camtepe, S., & Atiquzzaman, M. (2019). Local differential privacy for deep learning. IEEE Internet of Things Journal, 7(7), 5827-5842.
> >
> > 2. In your pipeline, the output is in the trusted zone. This surely can protect membership inference attack. But it's not the membership inference attack in HE. A well-held truth is that, MIA can still work if the adversary knows the decrypted data in HE. Therefore, MIA can not be mitigated by HE. If you say that your protection can mitigate MIA, then all HE can also protect MIA. I don't think it's a proper way to consider MIA in HE.
> >
> >
> > I won’t downgrade the attack and evaluation contributions—they are solid. I will retain my current score.

---

> ### Author Response · Authors · 2025-11-27
>
> Thank you for your timely response to our comments.
>
> **Comment 1: "Consider retitling or clarifying in the abstract that the primary contribution is threat discovery and evaluation, not a production-ready privacy mechanism."**
>
> Please note that the revised paper has already been retitled to "Securing Transfer Learning: Few-Shot Training-Data Reconstruction Attacks and Reverse-Role Homomorphic Encryption". We will make further revisions to improve the paper's balance.
>
> We do believe that all the main contributions listed in points 1-5 in the Introduction are equally important. These points mention the attacks, the evalutaion as well as the RHE mechanism.
>
> **Comment 2: "MIA cannot be mitigated by HE in conventional settings."**
>
> We thank the reviewer for raising this point. MIA in the context of HE is a well-studied problem, and we agree that the key issue is who gets to see decrypted outputs. Please note that our hard-label black-box reconstruction attacks prove that an adversary having access to model's decisions can indeed perform a successful training data reconstruction attack (not just MIA). This motivates the necessity for model output encryption in RHE.
>
> In our work, **the adversary is the untrusted environment that hosts the model** (e.g. an outsourced compute provider). The owner of the TL training data, who holds the decryption key, is treated as a trusted party and already knows the training data. Under RHE, the adversary never observes either i) the TL-dependent head weights in plaintext, or ii) any decrypted logits or class decisions -- adversary can access only public HE parameters and ciphertexts that are secure.
>
> Note that our precise threat model arises in many practical situations when outsourcing inference for models trained on private/regulated data, for instance:
> * medical models deployed in the cloud (a hospital trains a head-NN on its own small, sensitive patient dataset),
> * financial risk scoring and fraud detection (bank trains a head-NN on its own transaction history, per-customer credit data, or fraud labels),
> * biometrics and access control (an organization trains a head-NN for their employees' face recognition, speaker verification)
> * public social media content moderation where the TL training uses private data (but the model is queried on public data).
>
> We fully agree with the reviewer that if the adversary had access to decrypted outputs, then HE alone would not mitigate MIA (MIAs using only NN input-output pairs have been well-studied, also in the context of HE, see for instance in [SCC21]). However, **our claims do not rely on such a setting**. What our RHE construction contributes, compared to conventional HE deployments, is a concrete key-management and encryption layout that allows outsourcing inference to an untrusted environment while ensuring that this environment never sees the signals (weights or outputs) required for training-data reconstruction or membership inference and that prediciton latency is not hugely affected by the encryption.
>
> To clarify this aspect better, we will revise the manuscript to state more explicitly that **our guarantees are with respect to the untrusted host adversary in this threat model, and not against arbitrary parties who have decryption keys and/or full access to plaintext outputs**.
>
> [SCC21] Shin, J.; Choi, S.-H.; Choi, Y.-H. Is Homomorphic Encryption-Based Deep Learning Secure Enough? Sensors 2021

---

### Author Response · Authors · 2025-12-01
**Summary of the Rebuttal Discussion**

Our rebuttal discussion has the following clear aspects: reviewers agree that (i) we expose a serious and under-explored privacy threat for few-shot transfer learning, (ii) our reconstruction attacks and evaluation are technically sound and novel, and (iii) Reverse-Role Homomorphic Encryption (RHE) is a practical, well-motivated defense for this setting. The remaining issues were about framing and clarity rather than correctness, and we have revised the paper accordingly.

The discussion strengthened the paper and did not point to any critical flaws.

Main concerns raised by the reviewers:

C1. Clarify positioning, title, and threat model.

C2. Justify that DP-SGD comparison is fair and robust.

C3. Make RHE’s role and limitations precise.

C4. Paper’s generalizability to full-model fine-tuning and relation to practice.

C5. Clarify attack novelty and readability.


**Pre-rebuttal scores, discussion status and main concerns raised:**

* Reviewer **KKYk: 4 (discussion in progress), raised C1, C3, C5**. There were three significant misunderstandings of: i) the attack sophistication, ii) the need for a formal proof of defence, iii) the role of the trusted party in RHE (the reviewer thought that MIA not being defended). After discussion, they accepted the attacks as "solid” rather than "trivial", with a clearer understanding that the paper’s main novelty is threat discovery plus a well-engineered defense for that threat. We clarified that formal proof for (R)HE is well-established and that in RHE the adversary is the untrusted environment that hosts the model. This directly addresses the reviewer’s concern about MIA under HE: we only claim protection against the untrusted host, not against arbitrary parties with decryption keys or access to unencrypted outputs. Here, the discussion was cut short by the leak before the reviewer could respond.
* Reviever **HBNF: 6 (awaiting initial response), raised C2, C4**. We have thoroughly addressed this reviewer's concerns. They questioned whether our negative result for DP-SGD might be due to weak tuning. We documented our tuning strategy and **added plots and experiments** showing we explored reasonable hyperparameters and longer training, yet still observed a large utility drop at realistic privacy levels. Reviewer YfZy accepted that DP-SGD performs poorly in this few-shot TL setting due to this inherent limitation, not a bad baseline. We also note that our attacks and RHE could extend to full-model fine-tuning, albeit at significantly higher compute cost.
* Reviewer **dMgT: 4 (awaiting initial response), raised C4, C5**. We have thoroughly addressed this reviewer’s concerns. They and others questioned whether the frozen backbone + trainable head setting is too narrow. We now explain that this setup is common in sensitive, small-data domains (e.g., medical imaging, biometrics), outline how our attacks and defense may extend to variants like full-model fine-tuning or PEFT, and add latency evidence from prior HE work (**new Fig. 6 and discussion**) to justify focusing on shallow encrypted heads, where RHE is practical and accurate. We explicitly state that our design protects only the encrypted part of the model and does not secure an unencrypted fine-tuned backbone (which should not be applied in RHE). Deeper encrypted models are clearly marked as future work.
* Reviewer **YfZy: 4 (positive conclusion to rebuttal), raised C3, C4, and partially C2, C1**. This reviewer said their concerns were fully addressed and praised our response to HBNF. They had initially seen the RHE part as somewhat disconnected and asked about comparisons to more complex PPI frameworks. We clarified that RHE is scheme-agnostic and built on widely supported methods that the ML community can adopt immediately with high accuracy and good prediction latency. Since we now clearly frame RHE as a practical, easily deployable solution, it is not positioned as competing directly with full end-to-end PPI systems.

**Summary of attack novelty**

Concerns were mainly about clarity and how we positioned novelty. We added a pipeline diagram (**new Fig. 2**) and clearly highlight that we are the first to show white-box and hard-label black-box instance-level training-data reconstruction under the weak adversary (no knowledge of gradients, logits, or auxilliary training points). This was enabled by our **key novel design elements**, including reconstructor-NN architecture and our softmin-based loss for learning to select the most vulnerable data instance per class and resolving permutation symmetries. The result is a clearer narrative that connects the attack and the motivation for RHE. Finally, we stress that our work exposes unprotectable vulnerabilities under a realistic threat model, underscoring the **urgent need for new defences such as our proposed use of RHE**. This contrasts with prior attacks that rely on idealised assumptions very unlikely to hold in most applications.

---

### Meta-Review · Area_Chair_dcJ5 · 2025-12-28

**Summary:**

KKYk: (1) Scope and framing (the paper is about defense, but most pages develop and measure the attack and the DP-SGD tradeoff). (2) Lack of `head-to-head evaluation under RHE. (3) Focus on a trivial problem (reconstruction threat)

HBNF: (1) question regarding the DP-SGD baseline comparison. (2) question regarding the generalizability to non-frozen paradigms.

dMgT: (1) need a systematic description of the research background. (2) Lack of detailed steps for the proposed attack process. Lack of novelty. (3) No generalizability of the assumption (only fine-tuning head neural network during transfer learning) in practical applications.

YfZy: (1) The HE component of the work is disconnected from the core contribution. (2) The performance evaluation in Table 3 lacks contextualization with other privacy-preserving framework. (3) Need better motivation for the few-shot regime. (4) Need better clarity in Sec 3.

The ratings of this paper are mixed. While the rebuttal addresses some reviewers' concerns, there are still many outstanding reviewers' concerns. Only one reviewer (YfZy) explicitly mentioned increasing the rating. Another reviewer (KKYk) explicitly mentioned retaining his/her score. While other two reviewers have not responded, they have outstanding concerns not well addressed in the rebuttal. It is unlikely that they will raise their ratings. Given all these, the decision is that the paper is not ready for ICLR in its current form. The authors are encouraged to revise the paper (taking into account of reviewers' comments) for resubmission.

**Reviewer Concerns:**

KKYk: the reviewer concerns are still outstanding (the reviewer responded and retained his/her score)

HBNF: (1) is addressed in the rebuttal. (2) is still outstanding.

dMgT: (1) and (2) are partially addressed in the rebuttal. (3) is still outstanding

YfZy: most concerns are addressed (as explicitly acknowledge by the reviewer)

**Reviewer Scores:**

One reviewer (KKYk) explicitly mentioned retaining the original score. One reviewer (YfZy) mentioned increasing the score 4->6. Other two reviewers have not responded. Based on the discussion, it is unlikely that they will change the scores.

---

### Decision · Program_Chairs · 2026-01-26

Reject